# Dissecting the enhancer gene regulatory network in early *Drosophila* spermatogenesis

Patrick van Nierop y Sanchez [1], Pallavi Santhi Sekhar [1], Kerem Yildirim [1], Tim Lange [1], Laura Zoe Kreplin [1], Vigneshwarr Muruga Boopathy [1], Stephanie Rosswag de Souza [1], Kim Dammer [1], David Ibberson[2], Qian Wang [1], Katrin Domsch [1], Anniek Stokkermans [3], Shubhanshu Pandey [1], Petra Kaspar[1], Rafael Martinez-Gallegos[1], Xuefan Gao [1], Aakriti Singh [1], Natalja Engel[4], Fillip Port[5], Michael Boutros[5], Josephine Bageritz [4] & Ingrid Lohmann [1] ✉

Cellular decision-making and tissue homeostasis are governed by transcriptional networks shaped by chromatin accessibility. Using single-nucleus multiomics, we jointly profile gene expression and chromatin accessibility in 10,335 cells from the *Drosophila* testis apical tip. This enables inference of 147 cell type-specific enhancer-gene regulons using SCENIC + . We functionally validate key transcription factors, including *ovo* and *klumpfuss*, known from other stem cell systems but not previously linked to spermatogenesis. CRISPR-mediated knockout reveals their essential roles in germline stem cell regulation, and we provide evidence that they co-regulate shared targets through overlapping enhancer elements. We further uncover a critical role for canonical Wnt signaling, with Pangolin/Tcf activating lineage-specific targets in the germline, soma, and niche. The Pan eRegulon links Wnt activity to cell adhesion, intercellular signaling and germline stem cell maintenance. Together, our study defines the enhancer-driven regulatory landscape of early spermatogenesis and reveals conserved, combinatorial mechanisms of niche-dependent stem cell control.

Spermatogenesis is a conserved and highly coordinated developmental process in which GSCs give rise to mature sperm through mitotic divisions, meiosis, and differentiation. In the *Drosophila* testis, germline stem cells (GSCs) and somatic cyst stem cells (CySCs) are maintained by signals from non-dividing hub cells within a spatially organized niche (Fig. 1a), enabling reciprocal communication and tightly coupled lineage progression[1–4]. This architecture makes the testis an ideal model to study how transcriptional regulation and intercellular signaling coordinate stem cell maintenance and differentiation.

Previous genetic and transcriptomic studies have identified key pathways involved in early testis regulation, including JAK/STAT, BMP, and Hedgehog. The JAK/STAT effector Stat92E promotes both CySC self-renewal and GSC adhesion[5], while BMP signaling via Mad supports GSC maintenance[5,6]. The Hedgehog effector Cubitus Interruptus (Ci) is

[1]Heidelberg University, Centre for Organismal Studies (COS) Heidelberg, Department of Developmental Biology and Cell Networks—Cluster of Excellence, Heidelberg, Germany. [2]Deep Sequencing Core Facility, BioQuant, Heidelberg University, Heidelberg, Germany. [3]Hubrecht Institute-KNAW, Utrecht, The Netherlands. [4]Heidelberg University, Centre for Organismal Studies (COS) Heidelberg, Department of Stem Cell Niche Heterogeneity, Heidelberg, Germany. [5]German Cancer Research Center (DKFZ), Division of Signaling and Functional Genomics and University of Heidelberg, Department of Cell and Molecular Biology, Heidelberg, Germany. ✉e-mail: ingrid.lohmann@cos.uni-heidelberg.de

required for CySC identity independently of JAK/STAT[7,8]. In the germline, a major transcriptional transition occurs as cells progress from mitotically dividing spermatogonia to differentiating spermatocytes. Spermatogonia undergo four rounds of transit-amplifying divisions, giving rise to successive 2-, 4-, 8-, and 16-cell cysts (Sp-2, Sp-4, Sp-8, Sp-16), before entering the meiotic phase. This transition culminates in the formation of primary spermatocytes (1 °Sc), which subsequently

give rise to secondary spermatocytes (2 °Sc) following the first meiotic division. During this transition, a profound transcriptional shift occurs, leading to the activation of over 1000 new genes in preparation for meiosis and spermatid differentiation[9–12]. This wave of transcription is coordinated by the Meiosis Arrest Complex (tMAC), testis-specific TAFs, spermatocyte-specific TFIID homologs, and the Mediator complex[13–22], which primarily target promoters of genes essential for

**Fig. 1 | Single-nucleus RNA-seq analysis identifies early germline and somatic populations in the _Drosophila_ testis. a** Schematic of an adult _Drosophila_ testis depicting germline (red) and somatic (blue) lineages, with an inset highlighting the niche and early progenitor cells. **b** UMAP embedding comparing our data (gray) with the Fly Cell Atlas (colored clusters) reveals high concordance across datasets. **c** Transcriptomic UMAP, illustrating two distinct trajectories representing germline and somatic differentiation; cells are colored by latent time. **d** Heatmap showing dynamic expression of germline DEGs across latent time, scaled per gene; marker genes, cell numbers, and transcriptional dynamics are indicated. **e** Equivalent latent time analysis for the somatic lineage. **f** Latent time-based clustering recapitulates

domains of known differentiation markers in transcriptomic space. **g** Matrixplot of established and candidate markers across clusters. **h**–**k** Validation of marker gene expression using _vasa_[EGFP] (magenta) testes to visualize the germline, hub cells are delineated by E-cad or Fas3 (yellow) and somatic cells by Tj (yellow). **h** _ovo_ mRNA (green) localizes to Vasa-positive germline cells (white dashed lines) adjacent to hub cells (Ecad, yellow). **i** Org-1 protein (green) is restricted to hub cells. **j** _DWnt6_ mRNA (green) is detected in CySCs (blue dashed lines) and hub cells (white dashed line). **k** _tj_ mRNA (green) colocalizes with Tj in early somatic cells. Scale bars, 15 μm. See also Supplementary Fig. 1, Supplementary Tables 1 and 2, Supplementary Data 1–4.

late-stage germline differentiation[23]. Germline differentiation is tightly coordinated with dynamic changes in the somatic support cells: early spermatogonia are encapsulated by early somatic cyst cells (eSCCs), which transition into late somatic cyst cells (lSCCs) as development proceeds. Subsequently, the somatic lineage diverges into two transcriptionally distinct branches—somatic differentiation branch A (SdA) and branch B (SdB)—which likely support specific germline stages[9]. This synchronized co-differentiation highlights the functional interdependence of the germline and soma within the testis niche.

Despite these insights, it remains unclear how signaling pathways and transcriptional regulators converge at the level of chromatin to drive lineage-specific gene expression programs. In particular, the role of regulatory elements orchestrating developmental transitions in the testis have not been systematically mapped. Enhancer-driven transcriptional networks, or eRegulons, composed of TFs, accessible regulatory elements, and their target genes, provide a framework to decode such logic[24–26]. While single-cell multi-omic studies in the _Drosophila_ brain have successfully reconstructed eRegulons[27], comparable datasets for the testis are lacking, and early progenitor populations are underrepresented in existing atlases[9,11,28].

Here, we apply single-nucleus multi-omics to jointly profile gene expression and chromatin accessibility in over 10,000 early testis cells. Using SCENIC+, we reconstruct 147 high-confidence eRegulons, validate key TFs in vivo, and identify extensive predicted co-regulation of target genes across germline and somatic lineages. We identify canonical Wnt signaling as a key regulator of GSC number, with the effector TF Pangolin/Tcf predicted to target genes in GSCs, CySCs, and hub cells - predictions we validated in GSCs. Our data define the enhancer logic underlying stem cell function in the testis and are made accessible via an interactive web application (dAWA), providing a resource for dissecting transcriptional and signaling networks during spermatogenesis. Our work establishes a foundational framework for understanding how enhancer-mediated regulation orchestrates stem cell dynamics and signaling crosstalk during tissue development.

## Results

### Multimodal profiling reveals rare cells in _Drosophila_ tests
To investigate regulatory programs in the _Drosophila_ testis, we generated a single-nucleus multi-omics atlas by combining RNA expression and chromatin accessibility (snRNA-seq and snATAC-seq) from cells at the apical tip of adult testes. This region contains the stem cell niche, which is essential for understanding stem cell regulation and early differentiation (Fig. 1a and Supplementary Fig. 1a). To enrich underrepresented niche cells in existing single-cell datasets[9,11,28], we microdissected tissue, isolated nuclei, and used the 10x Genomics Multiome platform to jointly profile gene expression and chromatin accessibility per nucleus[29]. We profiled 10,335 high-quality nuclei across four replicates, capturing 13,034 protein-coding genes, 5839 non-coding RNAs, and 46,619 accessible regions. Compared to the Fly Cell Atlas, our dataset is enriched for early germline and somatic cells and depleted of terminally differentiated types, which comprise ~40% of FCA testis data (Fig. 1b and Supplementary Fig. 1d). Transcriptome-based UMAP embedding revealed clear separation of three cell groups: 7856 germline cells, 2428 somatic cells, and 51 hub cells (Fig. 1c).

Pseudotime inference and differential gene expression analysis (log2FC > 1.5 for germline, >3 for soma) revealed developmental progression consistent with previous studies[9,11]. In the germline, late spermatogonia undergo a major transcriptional shift, marked by increased gene expression and a transition to a more transcriptionally active state (Fig. 1d). Gene Ontology (GO) analysis of differentially expressed genes (DEGs) reveals a transition from signaling functions to biosynthesis and remodeling (Supplementary Fig. 1e), consistent with entry into the primary spermatocyte stage. In contrast, somatic differentiation is accompanied by a steady increase in gene expression (Fig. 1e, center right), which stands in contrast to the general trend of transcriptional downregulation during differentiation[30]. The most pronounced transcriptional changes occur at two inflection points - early and late somatic cyst cells (eSCCs and lSCCs) - characterized by widespread downregulation of genes relative to earlier stages (Fig. 1e, bottom right). GO analysis of somatic DEGs revealed enrichment for metabolic functions (Supplementary Fig. 1f), consistent with the role of somatic cells in supporting germline development[31,32].

To refine cell state classification and explore soma-germline coordination, we analyzed transcriptional kinetics using scVelo, which infers latent time from splicing dynamics[33]. Because germline and somatic cells differ in their transcriptomes and differentiation trajectories, splicing dynamics were computed separately. Cells were binned by stage, like GSCs, gonialblasts (Gbs), 2-cell to 16-cell spermatogonia (Sp-2 to Sp-16), excluding the 51 quiescent hub cells (Fig. 1f). Although no single marker gene was specific to early germline stages, combinations of _escargot_ (_esg_), _ovo_, _cyclin dependent kinase 4_ (_cdk4_) and _string_ (_stg_) enabled high-resolution classification (Fig. 1g). For example, the Snail-like TF gene _esg_, a stem cell marker in other systems[34] and known to control CySC maintenance[35], is expressed in GSCs, CySCs, and hub cells (Fig. 1f, g). _ovo_, a zinc-finger TF gene required throughout oocyte maturation[36,37], peaks in GSCs and Gbs (Fig. 1g, h), and declines in cells with increasing transcript levels of the differentiation factor encoding gene _bag-of-marbles_ (_bam_)[38] (Fig. 1f, g). The expression of _cdk4_ in mitotically dividing spermatogonia persists until the peak expression of _bam_ (4/8-cell spermatogonia, Sp-4/8)[39] protein-tyrosine-phosphatase encoding gene _stg_ remains in all mitotically dividing cell stages (Fig. 1f, g) up to stage-16 spermatogonia (Sp-16), where the cells undergo four rounds of transit-amplifying mitotic divisions. We validated marker expression patterns in vivo by single-molecule fluorescence in situ hybridization (smFISH)[40]. As predicted from transcriptomic data, _ovo_ is largely restricted to GSCs and Gbs, despite earlier assumptions of ovary specificity[41,42]. We also identified _org-1_ as a novel marker of hub identity (Fig. 1i), and verified expression of known (_traffic jam_) and novel (_DWnt6_) somatic markers (Fig. 1j, k).

Together, this dataset resolves rare early populations and enables precise discrimination of closely related cell states in both lineages, in particular in the germline. It recapitulates and extends prior transcriptomic findings and establishes marker gene combinations to molecularly define GSCs, Gbs, and early progenitor cells.

### Regulatory region features in _Drosophila_ testis
To align chromatin accessibility with transcriptional dynamics, we leveraged shared barcodes from the 10x Multiome platform to transfer

latent time assignments from snRNA-seq to snATAC-seq data, bypassing gene activity imputation and enabling a unified developmental trajectory (Fig. 2a). This integration facilitated downstream analyses of chromatin accessibility, predictive eRegulon inference, and germline-soma signaling predictions.

We defined 25 transcriptionally distinct clusters and identified 29,989 non-overlapping accessible chromatin peaks using MACS2[43], composing of 46,619 consensus-accessible regions (500 bp each) as calculated by pycisTopic[44], capturing chromatin dynamics across germline and somatic lineages. As expected from the extensive DNA remodeling characteristic of germline development[45], accessibility was higher in the germline (34,837 vs. 29,085 regions) compared to the soma, peaking in stage-16 spermatogonia (Sp-16) before declining with spermatocyte differentiation (Fig. 2c). In contrast, accessibility increased in the somatic lineage accompanying the secondary spermatocyte (2°Sc) stage (Fig. 2c), mirroring transcriptomic activity (Fig. 1d, e). This may indicate a shift in the regulatory interplay between germline and soma, coinciding with increased expression of metabolic genes (Supplementary Fig. 1f). Indeed, transcriptome data indicate higher transcriptional activity in somatic cells downstream of the branching point relative to germline cells at comparable stages[9,11].

Promoter regions (−1 kb to +100 bp from TSS) accounted for most accessibility[44], particularly in spermatogonia and primary spermatocytes (1°Sc), while intronic accessibility was higher in soma (Fig. 2d). The distribution across exonic, intronic, promoter, and intergenic loci was broadly similar between lineages, with low intergenic representation reflecting the compact *Drosophila* genome. Most genes were associated with multiple accessible regions by correlative analysis of SCENIC+[44] (Fig. 2d, see "Methods"), consistent with the presence of putative cis-regulatory elements, such as enhancers, contributing to gene regulation[27,46]. Promoter accessibility correlated with gene expression: for example, *zfh1*, a CySC marker essential for CySC maintenance and non-autonomous GSC support[47], showed high TSS accessibility in CySCs, which declined during differentiation alongside transcript levels (Fig. 2e). A similar pattern was observed for *stg* in the germline (Fig. 2f). Notably, *zfh1* accessibility declined more slowly than Zfh1 protein levels, suggesting that while Zfh1 protein becomes restricted to CySCs[47], the broader chromatin accessibility landscape persists transiently, potentially reflecting residual transcriptional competence in early differentiating cells.

To assess how chromatin state influences TF activity, we integrated expression with motif enrichment using pycisTarget[44], restricting analysis to accessible regions to increase specificity[48]. This revealed lineage- and stage-specific TF motif availability. For example, the Ovo motif was enriched in spermatogonia (Fig. 2g), implicating Ovo in germline regulation. Similarly, motifs for Mad, a Dpp/BMP effector required for GSC maintenance[6], and Klu, a regulator of progenitor commitment in the gut[49], were enriched in early germline cells. Interestingly, while *Mad* and *klu* transcripts remained high (Fig. 1g), motif enrichment declined, suggesting that chromatin closure may limit their regulatory activity during differentiation. Temporal motif analysis revealed dynamic TF motif accessibility across spermatogenesis. Motifs for Fru, Bab1, and Lov were enriched in primary spermatocytes (1°Sc), while Hth, Achi, and Vis were enriched in secondary spermatocytes (2°Sc) (Fig. 2g). Several of these TFs are implicated in germline differentiation[21,50–52], while others remain uncharacterized. In soma, motif enrichment changed more gradually (Fig. 2h): motifs for Stat92E, Tj, and Bowl−key regulators of early cyst cell development[5,53,54]−were enriched early and declined with differentiation.

In summary, by linking TF motifs to accessible regions within single nuclei, we reveal how regulatory potential shifts over time, consistent with stage-specific transcriptional programs and lineage progression. These findings would not have been possible by investigating the transcriptomics alone.

## eRegulons reveal cell-specific transcription regulators

To uncover the regulatory architecture of spermatogenesis, we integrated snRNA-seq and snATAC-seq data using the SCENIC+ framework[44] to construct a testis-specific enhancer gene regulatory network (eGRN). This network comprises modular eRegulons, each linking a TF to correlated accessible regions and predicted target genes at defined cell-stages (Fig. 3a and Supplementary Fig. 2a). TF-gene associations were inferred by expression correlation, refined by accessible TF motif presence within ±50 kb of targets. An illustrative example is the *ovo* eRegulon (Fig. 3b).

Although TF function is influenced by chromatin context and cofactors, most TF-DNA motifs are derived from in vitro data and lack context specificity. To address this, we integrated lineage-resolved differentially accessible regions (DARs), DEGs, and latent time to infer dynamic regulatory activity, revealing stage-specific targets of eRegulons such as *ovo* in GSCs, Gbs, and 2-cell spermatogonia (Sp-2) clusters (Fig. 3b, c). We identified 103 high-confidence activator eRegulons (Fig. 3c and Supplementary Fig. 2a) and 44 predicted repressors (Supplementary Fig. 2b, c). Activators grouped into germline-, soma-, or dual-lineage classes, while repressors were mostly lineage-specific, with *fru* as the sole exception, active in both somatic cells and 2°Scs. Several regulatory modules control transcriptional programs at specific stages: in early spermatogonia, *ovo*, *Chrac-14*, and *klu* eRegulons are active, suggesting previously unrecognized roles in early germline regulation. Notably, *ovo* also functions as a repressor in these cells (Supplementary Fig. 2b, c), consistent with its dual activity in the female germline[55]. As cells differentiate into primary and secondary spermatocytes (1°Scs, 2°Scs), eRegulon diversity increases. GO analysis using g:Profiler revealed enrichment for chromatin remodelers (*Chrac-14*, *BEAF-32*, *E(bx)*, *egg*, *nej*, *Clamp*, *Lam*, *pho*) and neuronal regulators (*pros*, *pdm3*, *nub*, *optix*, *Lim3*, *acj6*, *ab*, *ham*, *Dref*), consistent with known testis-brain parallels[56]. We also identified *achi* and *vis* eRegulons, likely mediating recruitment of tMAC components Always early (aly) and Cookie monster (comr) to activate genes required for meiotic progression[21]. In the somatic lineage, eRegulons primarily regulate signaling. Key TFs include *foxo*, *pointed (pnt)*, *pebbled (peb)*, *Eip75B*, *cubitus interruptus (ci)*, and *anterior open (aop)*. For example, *pnt* negatively regulates EGFR signaling during dorsal follicle cell patterning[57], and *ci* acts as the canonical Hh effector required for CySC self-renewal[8]. In hub cells, we detected a small set of highly specific eRegulons centered on *abd-A*, *foxo*, *ci*, *mirror (mirr)*, and *tj* (Supplementary Fig. 2b, c).

In summary, we identified context-specific regulators of spermatogenesis, with germline eRegulons enriched for chromatin and meiotic functions, and somatic modules for signaling pathways. All eRegulons are accessible via our Atlas Web Application (dAWA).

## Validating core eRegulon TFs in *Drosophila* testis

To functionally validate TFs predicted from eRegulon analysis, we performed cell type-specific CRISPR perturbations using the GAL4 > UAS system. All experiments were restricted to adulthood by shifting flies from 18 °C to 29 °C at eclosion to assess the role of the TFs during tissue homeostasis, age-matched controls were treated the same way (Fig. 4a). Activators were prioritized due to the higher false-positive rate among predicted repressors[48]. Lineage-specific drivers included *nos-GAL4,UAS-Cas9* for early germline interference[58] and *patched (ptc)-GAL4,UAS-Cas9* for early somatic lineages[3] (Supplementary Fig. 3a). TF depletion efficiency after CRISPR was confirmed via smFISH quantification of transcript levels in adult testes. Of nine TFs tested, six showed robust smFISH signals in controls that were markedly reduced upon CRISPR knockout (Fig. 4c, e, g, i and Supplementary Fig. 3c, e), and phenotypic analysis was limited to these. For two genes, RNAi based interference studies were performed in the somatic lineage using the *cS87-GAL4* driver. Phenotypes were quantified by GSC counts, total

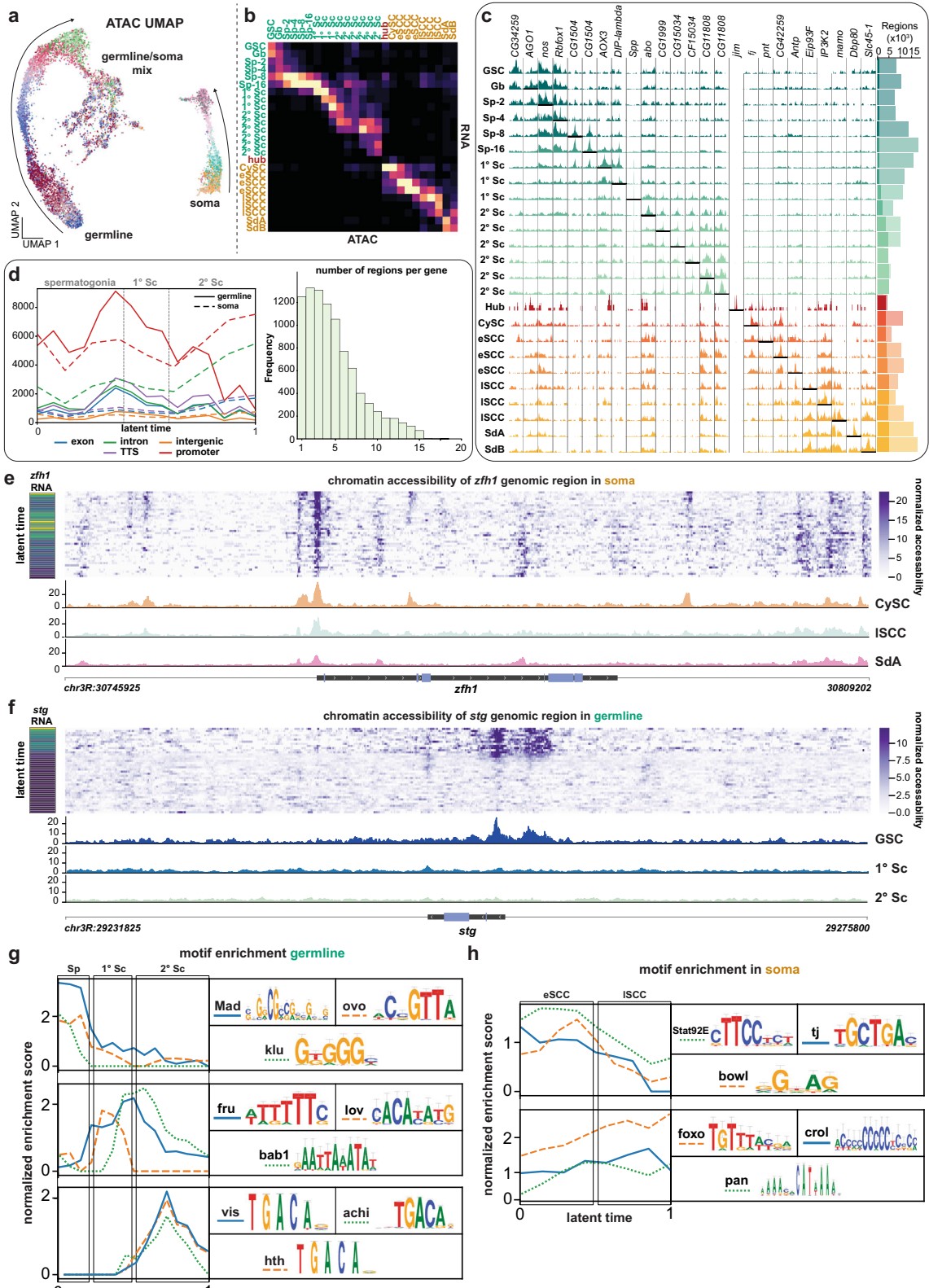

**Fig. 2 | Chromatin accessibility landscapes recapitulate lineage trajectories in the testis. a** UMAP embedding of single-nucleus ATAC-seq data, colored by transcriptome-based latent time clusters, shows distinct accessibility trajectories for germline and soma. (arrows). **b** Heatmap of label transfer scores between RNA and ATAC datasets. **c** Left: Representative cluster-specific accessible regions; underlined regions are cluster-enriched. Right: total and cluster-specific accessible regions per cluster. **d** Left: Distribution of accessible regions by genomic annotation; solid and dashed lines represent germline and somatic regions, respectively. Right: distribution of gene-associated peaks per gene. **e, f** Accessibility at the *zfh1* and *stg* loci ±20 kb, with cluster-specific tracks and temporal gene expression (insets). **g, h** Motif enrichment scores (normalized enrichment score, NES) across latent time for population-specific TFs in germline (**g**) and soma (**h**), identified via SCENIC+; logos indicate enriched motifs.

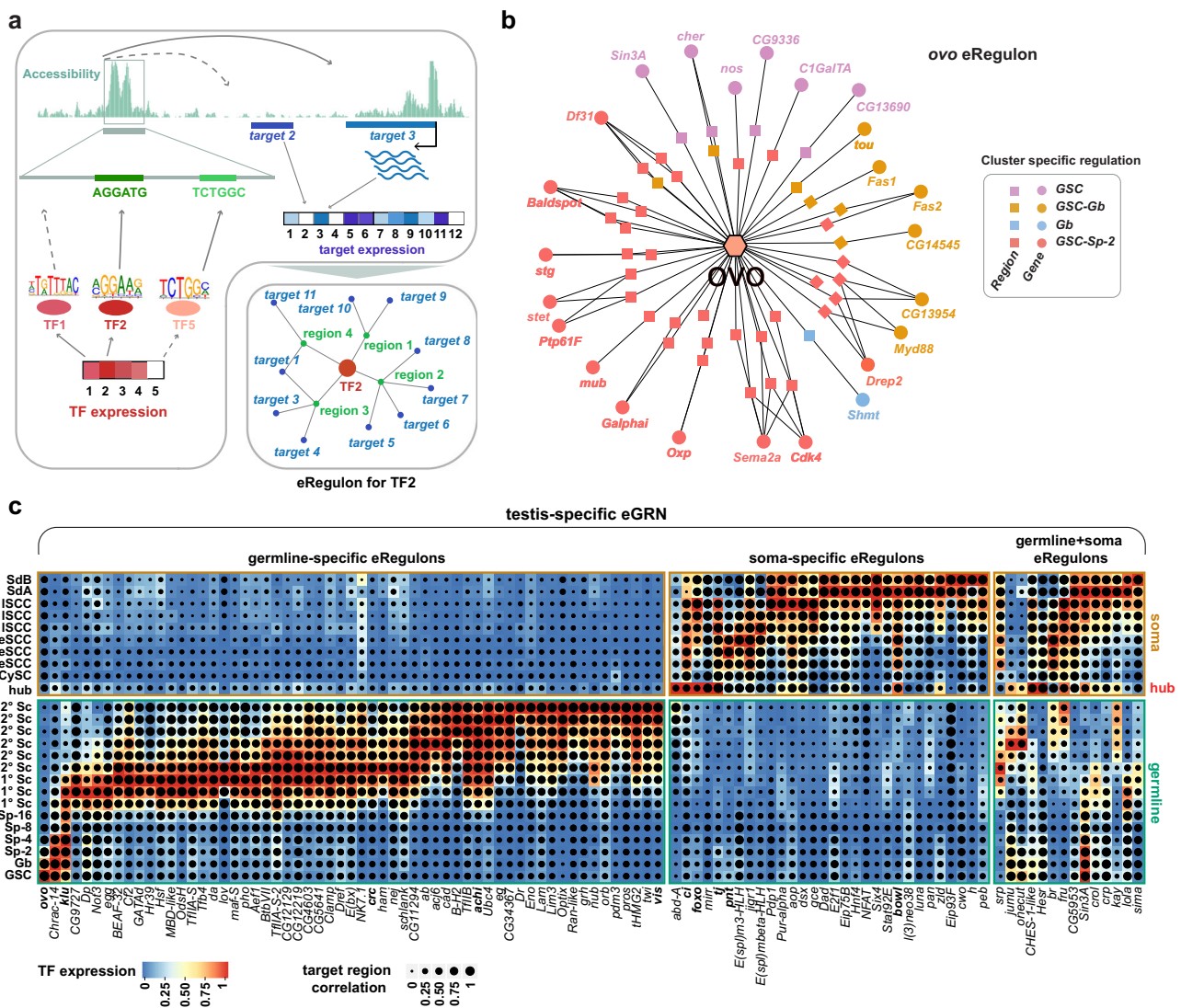

**Fig. 3 | Enhancer regulons (eRegulons) reveal TF-driven gene regulatory programs. a** Schematic of SCENIC+-based enhancer gene regulatory network (eGRN) inference. Regulatory links are defined by correlated TF expression, motif accessibility, and target gene expression. **b** Example *ovo* eRegulon with cell stage-specific TF-peak-gene associations in early germline clusters; colors indicate cluster-specific regulatory activity. **c** Dotplot heatmap showing TF expression (color) and correlation to accessible targets (dot size) across cell types. Lineage-specific eRegulons are ordered by maximal TF expression. See also Supplementary Fig. 2.

germ cell numbers, and GSC cell diameter in the apical testis of control and knockout animals (Fig. 4a; Source Data file).

To validate both the testis eRegulons and the CRISPR approach, we first targeted three known stem cell regulators: *zfh-1*, *Stat92E*, and *Mad*[5,6,47,59]. Knockout of *Mad* or *Stat92E* in the germline reduced GSC and total germ cell numbers (Fig. 4d and Supplementary Fig. 3h), consistent with prior reports of BMP and JAK-STAT pathway disruption leading to GSC loss[5,6]. Although previous *Mad* mutant studies showed complete germline loss[27], the phenotype here was milder, likely due to mosaicism from incomplete CRISPR efficiency. Residual *Mad* expression in germ cells and preserved expression in the soma (Fig. 4b, c) support this interpretation and confirm lineage specificity of the system. Somatic knockout of *zfh1* or *Stat92E* similarly reduced germ cell numbers (Supplementary Fig. 3d, f), in line with the role of somatic JAK-STAT signaling in GSC support[5,47,59]. Germline-specific *Stat92E* knockdown by RNAi testes with an accumulation of CySCs at the hub, recapitulating a previously described phenotype[59] (Supplementary Fig. 3g).

Having established the specificity and efficacy of CRISPR mutagenesis, we extended our analysis to additional eRegulon-predicted TFs with unknown testis function (Supplementary Table 1). Knockout of *cryptocephal* (*crc*), the *Drosophila* ortholog of *ATF4*, in the adult germline reduced GSC and germ cell numbers, and decreased GSC size (Fig. 4f), consistent with its role in sperm production in mice[60]. Given that *crc/ATF4* promotes hematopoietic stem cell maintenance via the integrated stress response (ISR)[61], these findings suggest a conserved role for ISR in germline homeostasis. Among the newly tested TFs, *ovo* and *klu* emerged as key regulators. Both are expressed in the early germline (Figs. 1g and 4b) and known to function in other stem cell systems[41,49,62]. While their expression domains partially overlap, *ovo* is restricted to GSCs and Gbs, whereas *klu* peaks at the 8 to 16-cell spermatogonial (Sp-8 to 16) stage (Fig. 4b). CRISPR knockout of either gene in adulthood led to striking defects: reduced GSC and germ cell numbers, and smaller GSCs (Fig. 4h, j), highlighting their essential roles in spermatogenesis. We also tested *tj*, identified by eRegulon analysis and commonly used as a somatic marker, but not further studied functionally in the testis. Adult-specific, soma-targeted *tj* knockdown, which reduced Tj protein levels in the testis (Supplementary Fig. 3i), significantly reduced GSC and germ cell numbers without strongly affecting GSC size (Supplementary Fig. 3i).

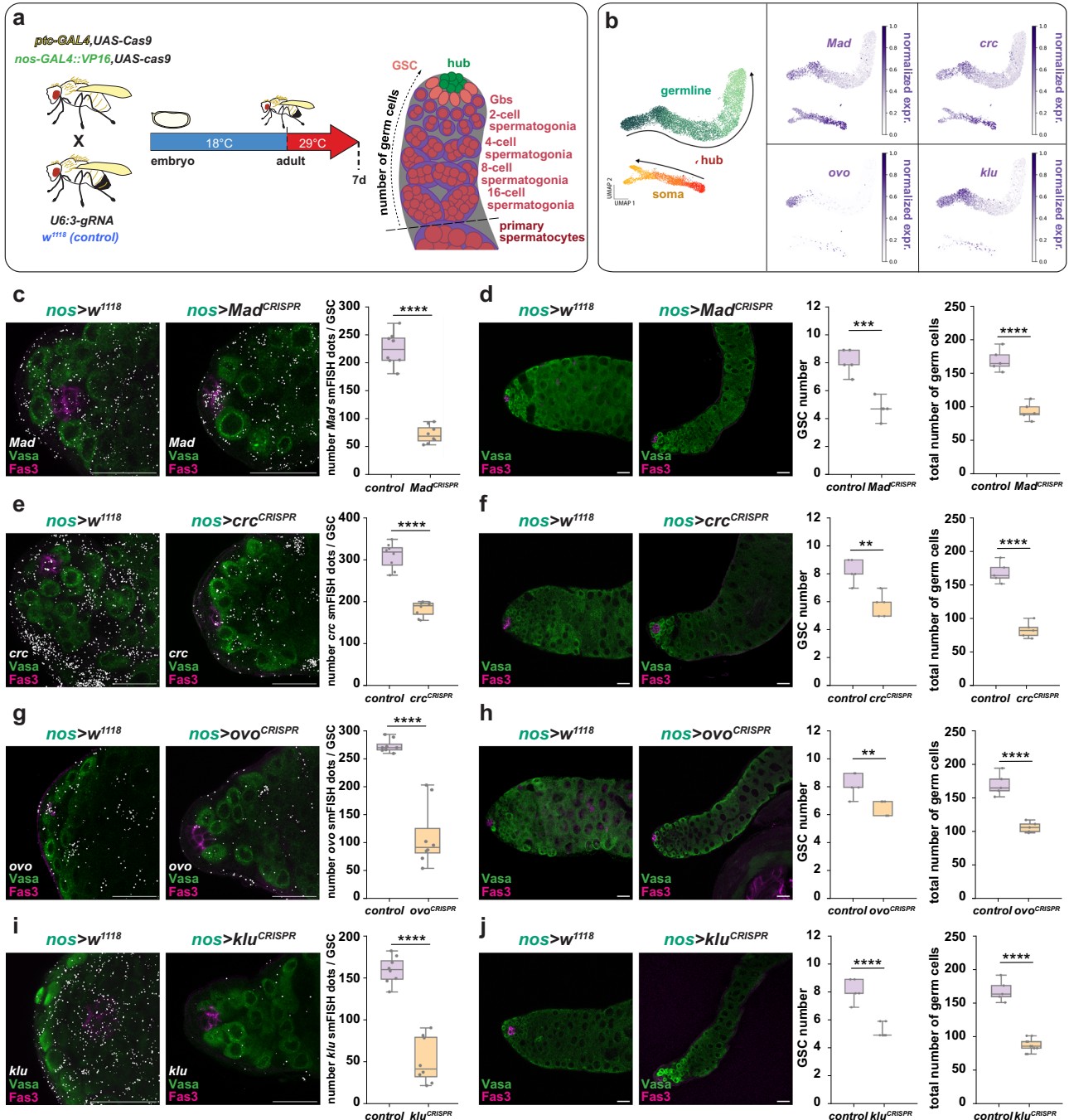

**Fig. 4 | Cell type-specific depletion of eRegulon TFs in the adult testis reveals functional roles in stem cell regulation. a** Schematic of the experimental strategy for TF validation. Lineage-specific GAL4 drivers (carrying *UAS-Cas9*) were crossed to U6:gRNA lines targeting selected TFs. Crosses were maintained at 18 °C until eclosion, then shifted to 29 °C to induce GAL4-driven expression. **b** Left: UMAP embedding of the dataset in transcriptional space, with cells colored by lineage-specific latent time. Right: Expression of indicated genes projected onto the UMAP; expression levels are shown as increasing intensity of purple. **c** Left: Representative smFISH images of *Mad* transcripts in control and germline-specific *Mad* knockout (KO) testes. Right: Quantification of *Mad* smFISH signal across genotypes. **d** Left: Representative images of control and germline-specific *Mad* CRISPR mutagenesis testes. Right: Quantification of GSC number, GSC diameter, and total germ cell number. **e** Left: Representative smFISH images of *crc* in control and germline-specific *crc* CRISPR mutagenesis testes. Right: Quantification of *crc* smFISH signal. **f** Left: Control and *crc* CRISPR mutagenesis testes. Right: Quantification of GSC

number, GSC diameter, and total germ cell number. **g** Left: Representative smFISH images of *ovo* in control and germline-specific *ovo* CRISPR mutagenesis testes. Right: Quantification of *ovo* smFISH signal. **h** Left: Control and *ovo* CRISPR mutagenesis testes. Right: Quantification of GSC number, GSC diameter, and total germ cell number. **i** Left: Representative smFISH images of *klu* in control and germline-specific *klu* CRISPR mutagenesis testes. Right: Quantification of *klu* smFISH signal. **j** Left: Control and *klu* CRISPR mutagenesis testes. Right: Quantification of GSC number, GSC diameter, and total germ cell number. In all images, Vasa (green) labels the germline, Fas3 (magenta) marks hub cells, and smFISH signals are shown in white. Scale bars, 20 μm. (**c**, **e**, **g**, **i**: *n* = 8), (**d**, **f**, **h**, **j**: *n* = 5). Boxplots represent median values within the Q1–Q3 range, while whiskers include the 0–100 percentiles. *p* values are calculated with two-sided independent *t*-tests. (*\*p* < 0.05; \*\**p* < 0.01; \*\*\**p* < 0.001; \*\*\*\**p* < 0.0001). Exact *p* values can be found in the Source data file. See also Supplementary Fig. 3, Supplementary Tables 1, 2. Source data is provided as a Source Data file (Fig. 4).

Together, these findings confirm that eRegulon-predicted TFs, including *ovo*, *klu*, and *crc*, are required for adult testis homeostasis and demonstrate the predictive accuracy of our integrated snRNA-seq and snATAC-seq framework.

## Multi-modal analysis identifies TF co-regulatory activity

eRegulon predictions, supported by phenotypic similarities from TF perturbation studies, suggested that co-expressed TFs may act through shared regulatory programs. To test this, we quantified overlap in predicted regulatory regions and target genes across eRegulons (see "Methods"). Despite variable overlap, both somatic and germline eRegulons showed substantial convergence (Fig. 5a and Supplementary Fig. 4d), indicating that TFs may coordinate their regulatory function via shared cis-regulatory elements to drive lineage-specific expression programs.

A particularly illustrative example of such potential cooperation is provided by the TALE-class homeodomain TFs *achintya* (*achi*) and *vismay* (*vis*) in secondary spermatocytes (2 °Sc) (Figs. 5a and S5a). These TFs exhibit extensive eRegulon overlap, co-expression, and co-enriched motifs in open chromatin regions, and have been shown to physically interact[21]. Their interaction with the tMAC complex is essential for regulating spermatocyte-specific gene expression during meiosis and spermatid differentiation[21], providing a proof-of-concept that co-expression, motif co-occurrence, and eRegulon intersection can indeed reflect direct combinatorial regulation. GO term enrichment of shared Achi/Vis targets revealed functions in cellular reorganization, cilium motility, and metabolism (Supplementary Fig. 4c), characteristic of other meiotic arrest genes[13,19]. Motif analysis also identified enrichment for TFIIB in their shared regulatory regions (Supplementary Fig. 4b), consistent with the known promoter-proximal binding of Achi and Vis[63].

Encouraged by this, we examined *ovo* and *klu*, two TF genes expressed in overlapping early germline populations (Figs. 1g and 4a) and associated with strikingly similar phenotypes upon CRISPR perturbation (Fig. 4h, j). GO term analysis of their shared and unique eRegulon targets implicated both in the regulation of the cell cycle and stem cell differentiation (Fig. 5e). Shared targets included *nanos* (*nos*), *stg*, *toutatis* (*tou*), and *stem cell tumor* (*stet*) (Fig. 5b), all functionally linked to germline or ovarian stem cell maintenance[1-4,64-66]. These genes were predominantly expressed in early germline populations (Fig. 5c), consistent with the expression domains of *ovo* and *klu*. Genomic regions associated with their shared targets were also enriched for the binding motif of Lola (Fig. 5d), a TF known to be required for GSC maintenance[67]. To test whether *ovo* and *klu* regulate these shared targets, we performed smFISH for *nos* and *stg* following germline-specific, adult-only CRISPR knockout of each TF. Both genes were significantly reduced upon adult-stage knockout of *ovo* or *klu* in comparison to age-matched control testes (Fig. 5f, g), consistent with their predicted roles in regulating a shared gene module in early germ cells important for GSC maintenance.

Together, these findings suggest that *ovo* and *klu* likely participate in a shared transcriptional program critical for early germline function. The interaction between *achi* and *vis* illustrates how TF co-expression, motif co-enrichment, and eRegulon overlap can signal physical interaction and functional co-regulation. More broadly, our multimodal single-cell analysis maps of lineage-specific regulatory networks reveal candidate co-regulatory partnerships that may ensure the stability and precision of stem cell gene expression programs.

## Ligand-receptor predictions identify Wnt signaling

Numerous signaling pathways regulating stem cell behavior in the *Drosophila* testis have been well characterized[1,2,4,35,68-71], yet a comprehensive view of cell-cell communication remains incomplete. To address this, we used our high-resolution single-nucleus transcriptome dataset to predict ligand–receptor interactions across testis cell types.

We applied the LIANA+ package[72] with a curated list of *Drosophila* ligand-receptor pairs[73] to infer signaling networks from gene expression data. This revealed both established and underexplored pathways active in early adult testes, particularly among hub cells, GSCs, and CySCs (Fig. 6a and Supplementary Fig. 5a). For instance, consistent with previous studies[7], we detected *hh* in hub cells and its receptor *patched (ptc)* in CySCs, confirming somatic Hh–Patched signaling (Fig. 6a). Likewise, *gbb* expression in hub cells and its receptor *tkv* in GSCs supports earlier findings that BMP signaling promotes GSC maintenance[6]. Notably, our analysis also uncovered less-characterized pathways, including Wnt signaling (Fig. 6a and Supplementary Fig. 5a), a central regulator of stem cell function in other tissues such as the ovary[74-80], but not yet systematically studied in the testis[81].

To validate Wnt pathway predictions from our transcriptomic analysis, we used smFISH to visualize mRNA expression of Wnt components in early testis cell types. Consistent with the snRNA-seq data (Fig. 6b), smFISH confirmed strong *DWnt4* and *DWnt6* expression in hub cells and CySCs, but little to no signal in germline cells (Fig. 6c). In contrast, components with lower transcript levels in the snRNA-seq data showed variable smFISH signal intensity and spatial distribution. For example, *wg* and *DWnt5* were weakly expressed in hub cells and CySCs by both methods, whereas *fz* and *fz2*–the most highly expressed Wnt receptor genes in the niche according to snRNA-seq–exhibited weak but detectable smFISH signals in all niche cells, including GSCs (Fig. 6c). Similarly, the canonical co-receptor gene *arrow* (*arr*) and the non-canonical co-receptor gene *off-track* (*otk*) were broadly but weakly expressed in all niche cells by snRNA-seq (Fig. 6b), a pattern mirrored by smFISH (Fig. 6c), with *otk* showing relatively stronger expression in hub cells and CySCs.

To dissect the role of Wnt signaling in adult testis homeostasis, we used gene-specific UAS-RNAi lines driven by lineage-restricted GAL4 drivers (*nos-GAL4*[58] for germline, *c587-GAL4*[82,83] for the somatic lineage, *upd-GAL4*[84] for hub cells) under temporal control (*tub-GAL80ts*), enabling adult-specific knockdowns following temperature shift (Fig. 6d). Expression and interaction data identified hub cells and CySCs as key sources of Wnt ligands (Fig. 6a–c). Consistent with this, adult-specific *DWnt4* depletion in hub cells (*upd-GAL4;tub-GAL80ts*) significantly reduced GSC numbers, while *DWnt6* knockdown increased them (Fig. 6e, f). Strikingly, the reverse was observed when *DWnt4* or *DWnt6* were depleted in the somatic lineage including CySCs (*c587-GAL4;tub-GAL80ts*): *DWnt6* knockdown reduced GSCs, whereas *DWnt4* depletion increased them (Fig. 6i, j), suggesting ligand-specific, lineage-dependent effects on GSC maintenance. To test whether GSCs directly receive Wnt signals, we knocked down *fz*, *fz2*, *fz4*, *arr*, and *otk* in the germline using *nos-GAL4*. Adult-only interference significantly reduced GSC numbers (Fig. 6g, h), indicating that Wnt reception in GSCs is essential for their maintenance. Notably, smFISH revealed low *fz*, *otk*, and *arr* transcript levels in adult GSCs (Fig. 6c), showing that even low expression permits functional Wnt responsiveness and RNAi sensitivity, supported by previous findings[85]. Mild phenotypes from individual *fz* knockdowns further suggest receptor redundancy. We next asked whether Wnt reception in somatic cells influences GSC behavior. Knockdown of *fz*, the most strongly expressed Wnt receptor, in hub cells or the somatic lineage reduced (or increased) GSC numbers (Fig. 6e, f), indicating that Wnt acts on somatic support cells to modulate the niche environment. Similar phenotypes were observed after continuous developmental knockdown (Supplementary Fig. 5b–g) suggest that Wnt signaling primarily contributes to adult GSC homeostasis, though an additional developmental role cannot be fully excluded.

Together, these findings show that somatically derived Wnt ligands from hub cells and CySCs promote GSC maintenance by acting on both germline and soma. Canonical and non-canonical Wnt pathways contribute to this regulation, and even low receptor expression in GSCs ensures responsiveness and supports stem cell maintenance.

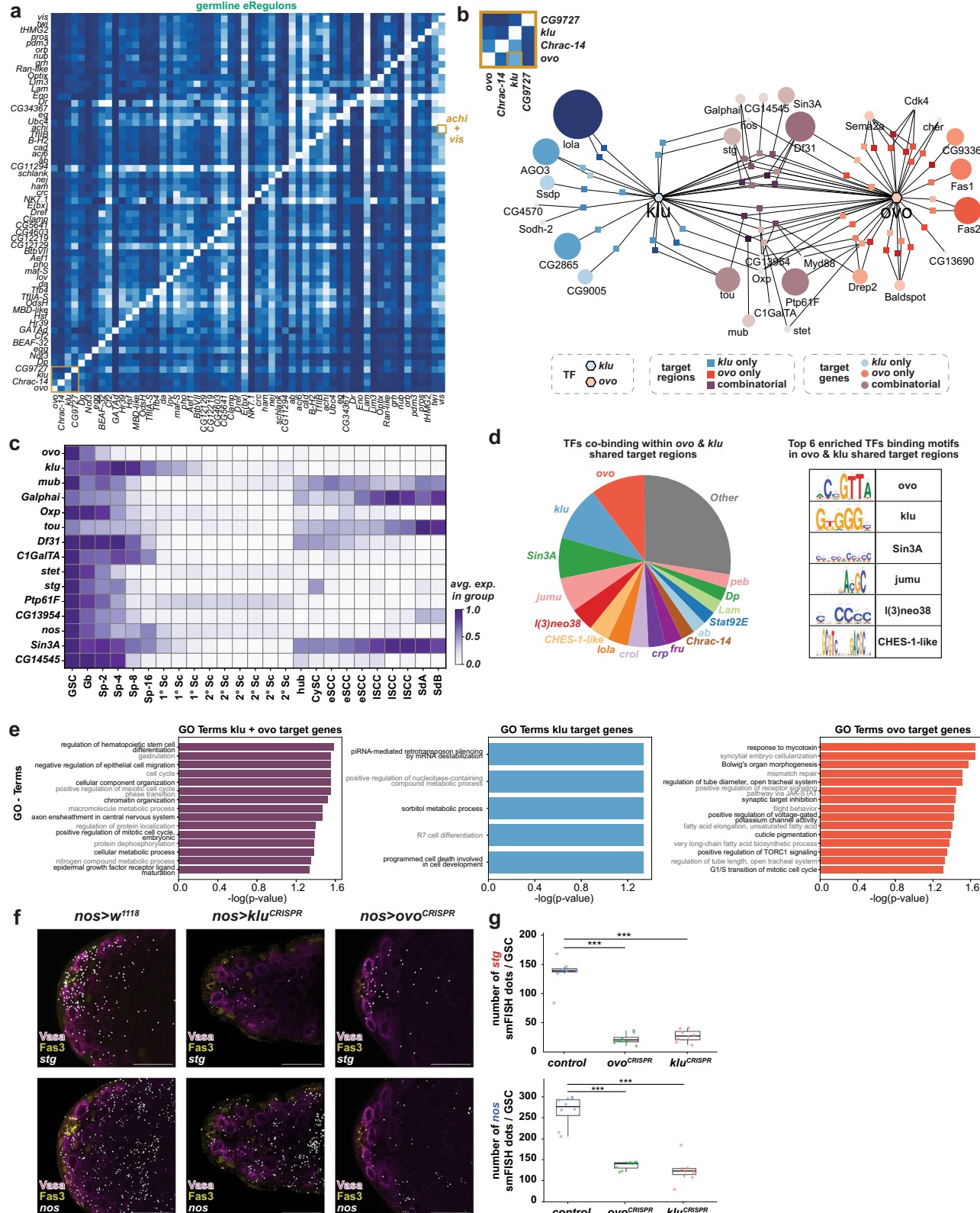

**Leveraging multimodal data to decode Wnt control in testis**

Functional perturbation of Wnt pathway components revealed a critical role for canonical Wnt signaling in GSC maintenance. This pathway requires the co-receptor Arr/LRP6, which stabilizes Armadillo (Arm) to promote its nuclear translocation and convert Pangolin (Pan; TCF/LEF homolog) from a repressor to an activator. A Wnt-responsive reporter driven by multimerized TCF motifs[86] was active in GSCs,

CySCs, and hub cells (Supplementary Fig. 6a), consistent with Pan eRegulon activity across these populations (Supplementary Fig. 6b, c, g). In GSCs, Pan is predicted to activate *how* and *lola*, two genes essential for GSC maintenance[67,87] (Supplementary Fig. 6b); in hub cells, it regulates adhesion-related genes including *shotgun* (*shg*), *Cad87A*, and *gliolectin* (*glec*)[88] (Supplementary Fig. 6c); and in CySCs, it targets genes enriched for signaling functions (Supplementary Fig. 6g).

**Fig. 5 | Overlapping eRegulons co-regulate distinct and shared gene sets.**
**a** Heatmap of normalized intersection scores among germline eRegulons based on shared regulatory regions; highlighting overlapping eRegulons in the early germline and the overlap of the *achi* and *vis* eRegulons. **b** *ovo* and *klu* eRegulons in GSCs illustrating uniquely and jointly regulated genes and regions. Inset shows the overlap between *ovo* and *klu* from (**a**). **c** Matrixplot of expression for candidate co-regulated genes. **d** Pie chart of TF motifs in *ovo-klu* shared regions; less frequent TFs grouped as *Other*. Table lists six top enriched motifs. **e** GO term analysis of uniquely and jointly regulated targets. **f** *stg* and *nos* expression visualized by smFISH in age-matched *nos > w^{1118}* (control), *nos > klu^{CRISPR}* and *nos > ovo^{CRISPR}* testes with CRISPR interference only performed in adulthood. **g** Quantification of *stg* and *nos* mRNA levels in the respective genetic backgrounds. (**g**: *n* = 8 for all experiments). Scale bars = 20 µm. Boxplots represent median values within the Q1–Q3 range, while whiskers include the 1.5*IQR percentiles. *p* values are calculated with two-sided independent *t*-tests. (**p* < 0.05; ***p* < 0.01; ****p* < 0.001; *****p* < 0.0001). Exact *p* values can be found in the Source data file. See also Supplementary Fig. 4, Supplementary Tables 1, 2 and Supplementary Data 4. Source data is provided as a Source data file (Fig. 5).

To assess functional relevance, we performed lineage-specific, adult-only RNAi against *pan*. Germline knockdown produced two phenotypes with equal frequency: in ~50% of testes, the hub was retained and GSC numbers remained comparable to controls, but total germ cell number was reduced (Fig. 7a–c); in the other half, the hub was lost and single germ cells were markedly increased (Fig. 7d, e). These findings suggest that *pan* function is required both to sustain germline progenitor expansion and to maintain niche architecture, with the observed variability likely reflecting differences in knockdown efficiency, timing of interference, or niche sensitivity to Wnt pathway perturbation. Notably, these phenotypes were more severe than those seen with hub-derived Wnt ligand depletion (Fig. 6e, f), likely due to (1) redundancy among Wnt ligands (e.g., *wg*, *DWnt4*) and (2) the dual role of Pan as a Wnt-dependent activator and repressor[89], making its loss more disruptive. Adult-specific knockdown of *pan* in hub cells did not alter GSC numbers after 7 days (Supplementary Fig. 6e–e″) but increased Fas3-positive (hub) cells by day 10 (Supplementary Fig. 6d, f, f'), indicating that hub cells actively respond to canonical Wnt signaling. This supports our predicted ligand-receptor interactions (Fig. 6a).

Given that Wnt pathway activation in somatic cells is essential for GSC maintenance, we next investigated how the expression of Wnt ligands themselves is regulated within the niche. To this end, we focused on *DWnt6*, a novel marker of CySCs. eRegulon analysis predicted regulation by nine TFs through 10 distinct regions, including Tj and Dsx (Fig. 7f). The Hh effector Ci was predicted to regulate *DWnt6* and *wg* in CySCs via a shared regulatory element (R6 + R7; Fig. 7g, i), and to regulate *DWnt6* in hub cells through the same region, which harbors additional motifs for Br, Tj, Dsx, and CG5953 (Fig. 7h). R6 + R7 is located near the *DWnt6* transcription start site and is accessible in CySCs and hub cells, but not GSCs (Fig. 7i). Reporter assays for the R6 + R7 region revealed strong enhancer activity in hub cells and weaker, yet detectable, expression in CySCs, consistent with chromatin accessibility patterns (Fig. 7i, j). Somatic knockdown of *ci* using *c587-GAL4* led to reduced expression of both *DWnt6* and *wg* in hub cells and CySCs (Fig. 7k, l), with *DWnt6* showing a more pronounced decrease, suggesting a greater dependence on Ci-mediated input. Together, these findings support a model in which Hh signaling promotes Wnt ligand expression and establishes a regulatory feedback loop between hub and CySCs (Supplementary Fig. 6h).

In summary, our multimodal single-cell analysis reveals a previously unrecognized regulatory circuit linking Hh and Wnt signaling in the *Drosophila* testis niche. By integrating chromatin accessibility, gene expression, and TF motif analysis, we identify upstream regulators of Wnt ligands and downstream targets of the effector Pan. These data uncover a reciprocal signaling axis between CySCs and hub cells that coordinates stem cell regulation across lineages. More broadly, this work demonstrates how multimodal inference can resolve intercellular signaling logic at single-cell resolution.

## Discussion

The adult *Drosophila* testis is a powerful model for dissecting both autonomous (cell-intrinsic) and non-autonomous (intercellular) gene regulation due to its spatially organized and continuously renewing stem cell niche. Using single-nucleus multi-omic sequencing, we generated a high-resolution map of chromatin accessibility and gene expression across early testis cell types. By integrating these data with TF binding motifs through the SCENIC+ framework[44], we reconstructed cell-type-specific regulatory networks and inferred intercellular interactions. This constitutes the first combined snATAC- and snRNA-seq dataset for the testis and is available to the community through an interactive web platform.

A major advance of this study is the ability to resolve transcriptionally similar cell types, such as GSCs and Gbs, which have remained challenging to distinguish[9,11,28]. This difficulty stems from the fact that the testis contains only two major lineages—germline and somatic—each undergoing continuous state transitions during differentiation. Our analysis shows that early spermatogonial stages cannot be uniquely defined by a single marker, but rather by precise combinations and expression levels of genes such as *esg*, *ovo*, *cdk4*, and *stg*. These subtle transcriptional shifts may underpin the capacity of early germ cells to dedifferentiate under physiological stress, such as during starvation-refeeding cycles[90–92].

To uncover gene regulatory logic within this dynamic system, we constructed extended eRegulons linking TFs to accessible chromatin regions and their predicted gene targets. These networks revealed stronger state specificity in the germline than in the soma, with distinct early- and late-acting regulatory modules. Among the early regulators, *ovo* and *klu* stood out for their prominent activity in GSCs and Gbs. *ovo* is a well-characterized regulator of maternal gene expression in the female germline[41] and modulates stem cell self-renewal in the intestine[62], while *klu* restricts enteroblast fate in the gut[49] and regulates neural stem cell proliferation[93]. Although both genes have known roles in other stem cell systems, their function in the testis had not been explored. Here, we show that CRISPR-mediated knockout of either gene disrupts testis function, and GO analysis of their predicted targets implicates them in stem cell differentiation, mitotic progression, and EGFR pathway regulation. These results extend the known regulatory scope of *ovo* and *klu*, identifying them as conserved factors that now include a role in early male germline development. Notably, while previous studies suggested that *ovo* is dispensable in males despite its expression in early germ cells of adult testes[41], our adult-specific knockout reveals a clear phenotype, indicating a functional requirement in the male germline. This finding is consistent with the work of Hayashi et al.[94], who showed that maternal *ovo-B* is essential in primordial germ cells of both sexes. The discrepancy likely reflects the use of hypomorphic alleles or RNAi in earlier studies, which may not have fully disrupted *ovo* function or captured temporally restricted phenotypes. Together, our results suggest that *ovo* plays a general role not only in germline development but also in the maintenance of germline identity and function during adulthood.

Where possible, we performed both RNAi and CRISPR-based perturbations for key TFs (Supplementary Table 1). Phenotypes were qualitatively consistent across methods, validating target specificity, but CRISPR consistently produced stronger effects. For example, RNAi against *ovo* resulted in no obvious phenotype, whereas knockout caused robust defects. This pattern, also observed in other systems[95], likely reflects incomplete knockdown and possible functional redundancy. Indeed, weak or absent phenotypes in some RNAi experiments may arise from compensatory regulation by other TFs. Supporting this, our analysis revealed combinatorial control within the same cells.

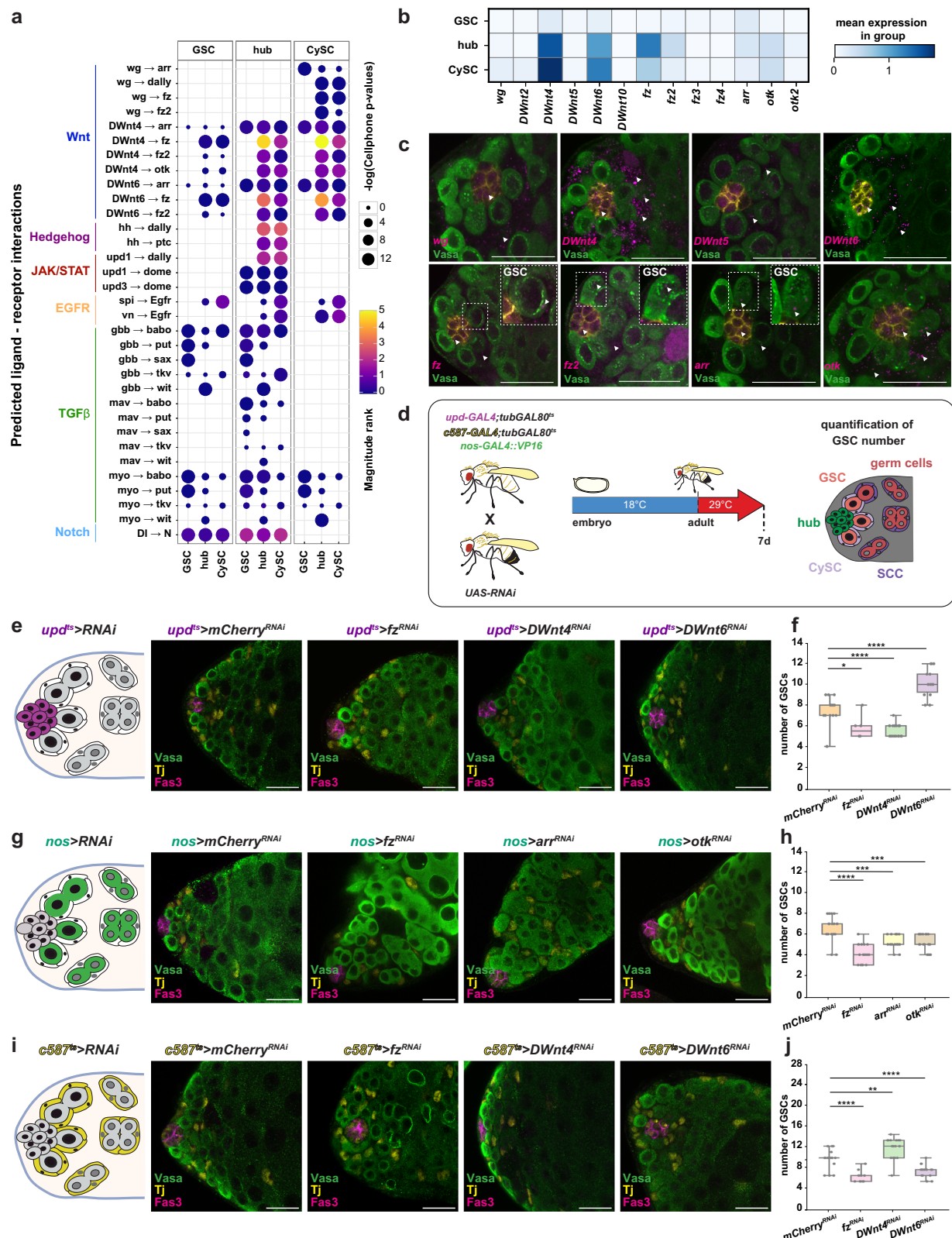

For example, Achi and Vis, which are known to form a protein complex[21,96], co-regulate a broad set of shared targets, whereas *ovo* and *klu* share only a focused group of highly significant targets, including *nos*, *stg*, *tou*, and *stet*. Some TFs, such as Ci, show minimal overlap with other eRegulons, suggesting independent activity or technical limits in detection. Further studies of regulatory element activity will help resolve these distinctions.

In addition to mapping cell-intrinsic regulatory networks, our dataset identifies active intercellular signaling pathways that shape cell identity and behavior. We uncover a key role for canonical Wnt signaling in maintaining GSC numbers, a function surprisingly overlooked despite prior identification of Wnt pathway components in the testis[81] and extensive work in the ovary[74,77–79]. A likely reason is that knockdown of individual Wnt ligands or receptors produced only mild

**Fig. 6 | Wnt pathway activity and function in the testis stem cell niche. a** Dotplot of inferred receptor-ligand interactions, grouped by pathway and cell type; dot size and color indicate statistical significance (−log *p* value) and interaction strength. **b** Matrixplot of mean expression of Wnt ligands and receptors in GSCs, CySCs, and hub cells. **c** smFISH validation of Wnt ligand and receptor gene expression in *vasa*[EGFP] testes. **d** Schematic of the lineage-specific RNAi approach to target Wnt pathway components. GAL4 driver lines were crossed to UAS-RNAi lines, with temporal control achieved by temperature shift in adulthood; testes were analyzed after 7 days in age-matched controls and knockdowns. RNAi-mediated knockdown of Wnt ligands or receptors in hub (*upd-GAL4;tub-GAL80*[ts]) (**e**), early germline (*nos-*

*GAL4*) (**g**), or soma (*c587-GAL4;tub-GAL80*[ts]) (**i**) affects GSC numbers. **f, h, j** Quantification of GSC numbers in the respective genotypes. Immunofluorescence markers: Fas3 (hub, magenta), Vasa (germline, green), Tj (soma, yellow). (**f**: *n* = 15, 6, 19, 14; **h**: *n* = 18, 19, 19, 17; **j**: *n* = 17, 15, 16, 19). Scale bars, 20 μm. Boxplots represent median values within the Q1–Q3 range, while whiskers include the 0–100 percentiles. *p* values are calculated with two-sided independent *t*-tests. (*$p < 0.05$; **$p < 0.01$; ***$p < 0.001$; ****$p < 0.0001$). Exact *p* values can be found in the Source data file. See also Supplementary Fig. 5, Supplementary Table 1, Supplementary Data 4. Source data is provided as a Source data file (Fig. 6).

---

phenotypes, likely due to functional redundancy[97]. In contrast, knockdown of the downstream effector *Pan/Tcf*, which integrates both transcriptional activation and repression, produced stronger and more variable effects, highlighting its central role in transducing Wnt inputs. Intriguingly, germline-specific knockdown of *pan* resulted in two distinct phenotypes: one with reduced germ cell numbers but preserved GSCs, and another with hub loss and dispersed germ cells. The former suggests a role for Pan in progeny expansion or early lineage progression, while the latter likely reflects secondary disruption of niche architecture due to impaired feedback or adhesion. The variability between phenotypes likely arises from differences in the timing or extent of *pan* depletion. Because Pan functions as both an activator and a repressor downstream of Wnt[89], its loss abolishes both regulatory arms of the pathway, explaining the broader and more severe effects observed. These findings establish *pan* as a critical regulator of germline homeostasis that couples lineage output with maintenance of niche structure.

Our study also underscores a key limitation of relying solely on transcriptomic data to infer gene function. For example, germline-specific RNAi of Wnt pathway components such as *fz*, *otk*, and *arr* produced strong phenotypes, even though their transcripts were barely detectable in GSCs by smFISH or snRNA-seq. This apparent discrepancy aligns with prior with findings by Horváthová et al.[85], who demonstrated that transcripts expressed below the detection threshold of single-cell or single-molecule assays can still be functionally relevant and subject to RNAi-mediated cleavage. Notably, genes involved in adhesion and signaling often operate at low expression levels, particularly in stem cells where tight spatial and temporal control is critical. These findings emphasize the need to complement transcriptomic profiling with functional assays to reveal essential regulators that may be sparsely expressed yet biologically indispensable.

In sum, our dataset represents the most comprehensive analysis of *Drosophila* testis stem cell stages to date, featuring single-nucleus transcriptomic profiles, chromatin accessibility data, gene regulatory networks, and signaling interactions. Accessible via a user-friendly web interface, this invaluable resource facilitates future studies aimed at unraveling the intricate genetic and signaling interactions within the testis. By providing detailed insights into both known and novel signaling interactions, our work advances the understanding of the regulatory networks that govern testis function and development. Comparing this dataset with other datasets, such as the multimodal dataset of the fly brain[27] or single-cell datasets of mammalian spermatogenesis[98], will be transformative. Future comparative analyses will uncover both shared and unique features, reshaping our understanding of conserved and divergent mechanisms in stem cell biology and development. These insights could transform how we study complex biological systems and drive major breakthroughs in the field.

## Methods
### Fly husbandry
Flies were fed on standard cornmeal food (cornmeal, barley malt, molasses, yeast extract, soy flour, propionic acid, Nipagin) and were

generally kept at 18 °C, 25 °C, or 29 °C. For testis nuclei isolation, *Drosophila* strain *w*[1118] was used like for the FCA dataset to make datasets comparable. Flies for testis nuclei isolation were raised at 25 °C, 60% humidity, and standard food. Virgin males were collected, kept at 25 °C, and the next day used for dissections. A table of all fly strains used can be found as Table S1.

### Nuclei preparation
Flies were anesthetized with $CO_2$ and their testes were dissected in ice cold 1x PBS. Apical tips were cut, aspirated to a new pre-coated microcentrifuge tube (1x PBS + 0.1% Triton X-100), flash-frozen in liquid nitrogen, and stored at −80 °C until further use. Each batch was completed within 10 min. The next day, all batches were thawed on ice, and nuclei were isolated by douncing 25 times with a tight pestle. Nuclei were counted on a C-Chip (DHC-N01) with Sytox Green as nuclear marker before loading. Detailed protocol is provided in document S1.

### snRNA and snATAC library generation
Library preparation was performed with 10x Genomics Single Cell Multiome ATAC + Gene Expression v1 chemistry, according to the 10x Genomics protocol (CG000338 Revision E), at the Deep Sequencing Core Facility Heidelberg. For library preparation, experiments 1 (exp1) and 2 (exp2) were performed in one batch, exp3 and exp4 were performed separately. For exp1 and exp2, 15k nuclei were loaded each. For experiments 3 and 4, around 50k nuclei were loaded each. Nuclei were transposed with Tn5 for 1 h at 37 °C, resulting in accessible DNA fragments with added adapter sequences. Following steps were according to the 10x Genomics protocol.

### Multiome library sequencing
Prior to sequencing, library fragment size was analyzed on a Bioanalyzer high-sensitivity chip. RNA and ATAC libraries were sequenced on separate flowcells. Libraries of exp1 and exp2 were sequenced jointly on an Illumina NextSeq550 High Output flow cell (GEX: 28-10-10-90 cycles, ATAC: 50-8-16-49 cycles) (exp1 barcodes: GEX SI-TT-A2, ATAC SI-NA-C1, exp2 barcodes: GEX SI-TT-B2, ATAC SI-NA-D1), exp3 on an Illumina NextSeq2000 P3 flow cell (GEX: 28-10-10-90 cycles, ATAC: 53-8-24-53 cycles), exp4 on an Illumina NovaSeq6000 S4 single lane (GEX: 101-10-10-101 cycles, ATAC: 101-8-24-101 cycles).

### Read alignment
The 10x genomics Cell Ranger ARC v2.0.2 pipeline was used in Cluster mode on a slurm-enabled system. The *D. melanogaster* assembly BDGP6.32, Ensemble release 107 was used as the reference genome, excluding non-primary scaffolds. The GTF file was filtered to include only protein coding genes and ncRNAs biotypes.

### RNA splicing matrices
Count matrices of premature (unspliced) and mature (spliced) abundances were obtained by running velocyto (v0.17.17) with default settings, using the *D. melanogaster* assembly BDGP6.32, Ensemble release 107 as reference genome.

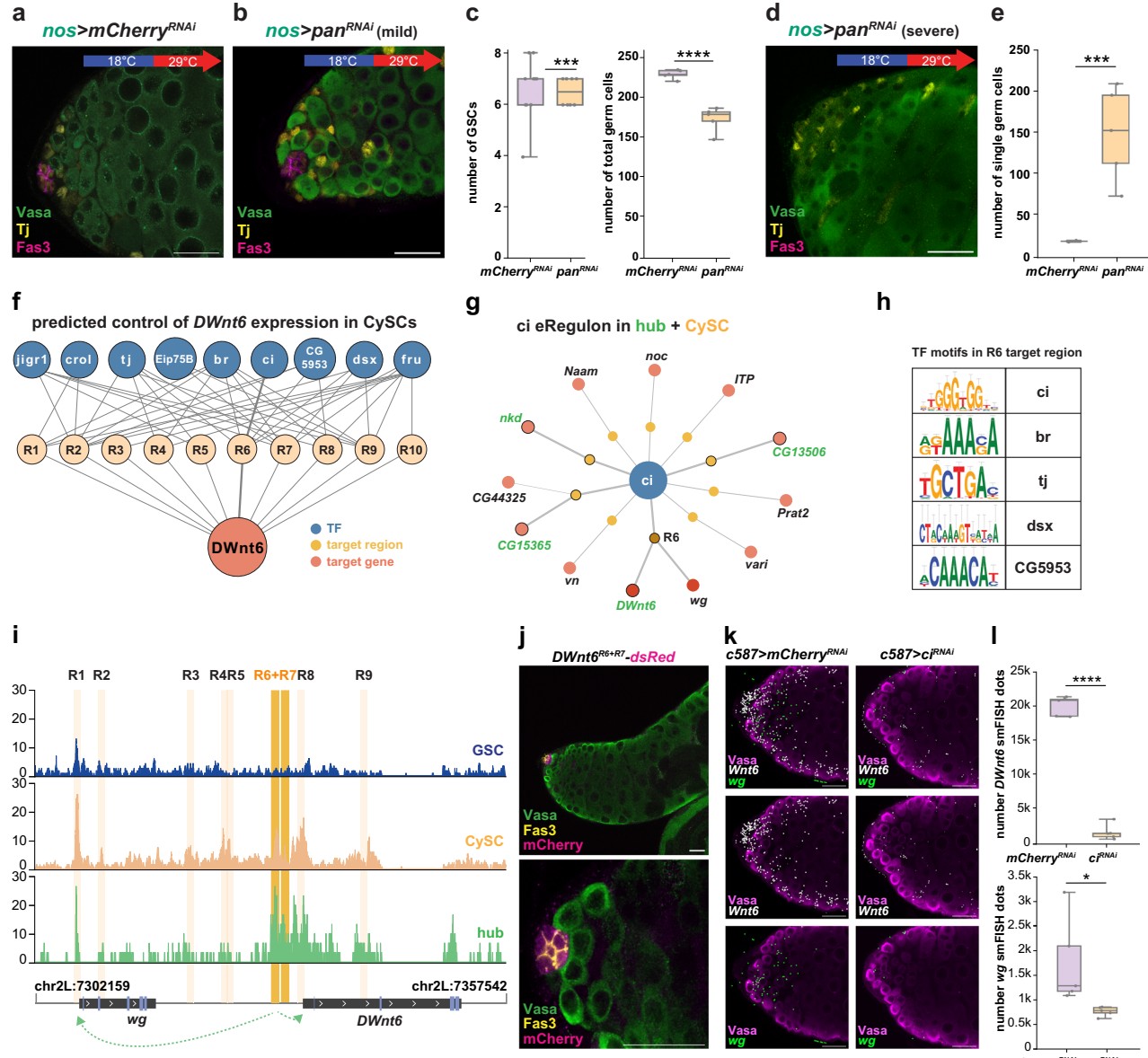

**Fig. 7 | Multimodal analysis identifies upstream and downstream regulation of Wnt signaling in the testis niche. a, b** Adult-specific knockdown of *pan* in the germline using *nos-GAL4* leads to a mild phenotype in 50% of testes. Immunostaining shows Fas3 (magenta, hub), Vasa (green, germline), and Tj (yellow, soma). **c** Quantification of GSC number and total germ cell number up to the spermatocyte stage in age-matched control and *pan*-RNAi testes. **d** Representative image of a severe phenotype upon adult-only germline-specific *pan* knockdown, showing hub loss. Immunostaining as in (**a**). **e** Quantification of individual germ cells up to the spermatocyte stage under control and experimental conditions. **f** TF network regulating *DWnt6* through 10 regulatory regions in CySCs. TFs are shown in blue, regions in orange, and *DWnt6* in red; bold lines highlight Ci binding at region R6 (chr2L:7330491-7330991). **g** *ci* eRegulon of in CySCs and hub cells. Putative target regions are shown in orange and their respective target genes in red. Targets shared between CySCs and hub cells (*DWnt6, CG15365, CG13506*) are circled. R6 is highlighted as a shared regulatory region for *wg* and *DWnt6*. **h** Enriched TF motifs in

regulatory region R6. **i** Chromatin accessibility in GSCs (blue), CySCs (orange), and hub cells (green). Gene bodies of *wg* and *DWnt6* are shown below, with exons in blue and transcriptional orientation indicated by arrows. Inferred regulatory regions are marked in orange. R6 and R7 form a single accessible regulatory region potentially regulating both *wg* and *DWnt6* (green arrows). **j** dsRed reporter driven by R6 + R7 region shows activity in somatic niche; Fas3 (yellow) marks the hub, Vasa (green) the germline, and mCherry (magenta) the reporter. **k** smFISH visualization of *Wnt6* (white) and *wg* (green) mRNA in *c578>mCherry^RNAi^* and *c587>ci^RNAi^* testes. **l** Quantification of *wg* and *DWnt6* mRNA levels under the indicated genetic conditions. (**c**: n = 18, 10, 5, 5; **e**: n = 5 for all; **l**: n = 5 for all). Scale bars, 20 μm. Boxplots represent median values within the Q1–Q3 range, while whiskers include the 0–100 percentiles. *p* values are calculated with two-sided independent *t*-tests. (*p < 0.05; **p < 0.01; ***p < 0.001; ****p < 0.0001) Exact *p* values can be found in the Source Data file. See also Supplementary Fig. 6, and Supplementary Tables 1, 2, Supplementary Data 4. Source data is provided as a Source Data file (Fig. 7).

### Annotation transfer from FCA transcriptome dataset

To calculate the global cell type enrichment scores, we integrated our GEX dataset with that of the FCA with harmonypy (v0.09). Scanpy's ingest was then used to transfer the annotation labels from the FCA to our dataset in UMAP embedding, with a knn = 50.

### snRNA-seq processing

The raw count matrices from the Cell Ranger Arc pipeline were used and subsetted according to the following thresholds: nuclei should contain a minimum of 100 genes expressed, have more than 1000 UMIs, and each gene should be detected in at least 3 nuclei. For the

number of genes and total UMI counts, nuclei containing less or more than the 1st and 99th percentiles were removed. Additionally, nuclei above the 99th percentile containing mitochondrial reads were also excluded from analysis. Resulting nuclei can be found in Supplementary Data 1. Nuclei doublets were removed per batch with the scrublet package[99] and the following parameters: doublet_rate=0.2, n_prin_comps=50, n_neighbors=15, threshold=0.3. Transcript levels were then normalized and log1p-scaled. Highly variable genes were calculated with scanpy.pp.highly_variable_genes and the following parameters: min_mean=0.0125, max_mean=3, min_disp=0.5. PCA was calculated with 50 components, and the transcriptomes of all batches were then integrated using harmonypy[100] (v0.09) and the previous pca embedding. Nearest 15 neighbors were calculated on the integrated transcriptome, and a leiden clustering resolution of 2 was applied. Next, cluster connectivity was calculated based on a partition-based graph abstraction (PAGA)[101], and was used to initialize the calculation of the UMAP embedding[102]. Clusters enriched for *cup* genes (spermatids), *IM4* (pigment cells), and clusters with maximum gene rank scores <40 were removed from further analysis, except for hub nuclei which were enriched in *Fas3*, *CadN*, and *org-1* (Supplementary Data 2). Resulting barcodes (Supplementary Data 3) were categorized into somatic, hub, and germline cells, and full transcriptomic dynamic modeling by scVelo (v0.3.0) was performed on both the somatic and germline separately. Root and end nodes were assigned by taking the barcodes with the maximum terminal score in the most distal clusters of each lineage. Resulting latent time was cut into 15 equally sized bins, to accommodate the spermatogonial stages based on marker genes, as explained in the main text. The time points of the somatic branches were combined to give two clusters in total, SdA and SdB nuclei.

### snATAC consensus regions generation
RNA latent time labels were transferred to the ATAC modality, and pseudobulk bed- and bigwig files were exported by these groupings. Narrow ATAC peaks were called with MACS2, with the following parameters: shift=73, extension size=146, q_value = 0.05. The consensus peak set was acquired by iteratively filtering out less significant peaks overlapping with a more significant one in 500 bp windows. Following on these consensus regions, between 5 and 100 topics and 150 iterations were calculated with the SCENIC+ model evaluation function. Based on the function's provided metrics, the optimal number of topics was set to 70. Experimental samples were integrated with harmonypy (v0.09).

### Calculation of DARs and expressed genes
DARs and DEGs were calculated as part of the SCENIC+ pipeline. Across all clusters, 14,577 DARs and 2008 DEGs (including ncRNAs) were identified.

### eGRN inference with SCENIC+
The eGRN was derived with the SCENIC+ pipeline (v1.0.1.dev1+-g3ec82fa) and default settings, except the regulatory motif search space which was set to 50 kb up and downstream of the TSS per gene. TF motifs were agglomerated by the S. Aerts lab into one motif collection database consisting of 32,765 unique motifs, containing motifs amongst others for 467 *Drosophila* TFs[44], and was used for this analysis. Analysis yielded an eGRN composed of 147 individual eRegulons, 103 activating and 44 repressing eRegulons.

### eRegulon target filtering and visualization
The eRegulon-based embedding was determined by calculating the per latent time cluster RSS values. RSS acts as a descriptor of the specific activity of an eRegulon with a value of 1 indicating exclusive activity in a cluster. Although per cluster ranking of RSS enabled us to identify the most specific eRegulons for a given cluster, or the cluster one eRegulon is most specifically active in, combination of the two was

not possible with the tools provided by SCENIC + . To enable a global comparison and sorting of eRegulons according to their specificity profile, we employed dimensionality reduction and clustering. Based on the per cluster RSS PCA was performed with the scanpy package using default settings, essentially treating individual eRegulons as cells and the latent time based clusters as genes (with the count values being composed of the per cluster RSS). Further KNN was computed and clusters were assigned based on the leiden algorithm[103]. Cluster based eRegulons were constructed based on the DARs and DEGs if not stated otherwise. Utilizing the NetworkX[104] package in python we supplied the eRegulons as a binarized graph to subsequently compute the layout based on the force directed Kamada-Kawai algorithm.

### GO term analysis
Enrichment analysis was performed by ranking gene expression per cell stage with scanpy and querying terms with the g:Profiler wrapper function. GO term redundancy was reduced by using the g:Profiler two-stage hybrid filtering system to highlight driver GO terms[105].

### Peak annotation with HOMER
Annotation of consensus peaks as TSS, exonic, intronic, promoter, and intergenic was performed by HOMER (v5.1) with default settings, TSS defaults to −1kb to +100 bp.

### Cell-cell communication predictions with LIANA+
Cell-cell interaction predictions based on ligand-receptor pairs were calculated from the gene expression modality with the LIANA+ package (v1.0.4)[72], using default parameters. The FlyPhoneDB resource provides a precompiled list of curated ligand-receptor pairs[73], and was used as input for the LIANA+ predictions.

### DataExplorer
Dash/Plotly was employed to interactively display the same plots as shown in the paper. Briefly, we use dash/plotly to write a html output natively in python. Using the interactive functionality of dash/plotly we enable the user to choose between different input options which mirror the analysis described by us. The interactive plots are displayed here as html-objects using the graph objects or px libraries. The code is available at https://github.com/Tim-Networks/dAWA.

### smFISH probe design
Probeset generation was performed with the Stellaris probe designer 4.2 online tool, https://www.biosearchtech.com/stellaris-designer. For each gene, part of the transcript covering all isoforms was chosen. Masking was set to level 5, which uses genomic information to increase specificity by reducing off targets. Probe length was 18-22nt, with a minimal spacing of 2nt. Probes were ordered from Eurofins Scientific pre-diluted to 200 pmol/μl in plates. A list of all the probes used can be found as Supplementary Data 4.

### smFISH probe generation
Probes were prepared using the enzymatic conjugation protocol (Gaspar et al.[43]). Amino-11-ddUTP (Lumiprobe, 15040) was conjugated with either NHS-esters of ATTO 550, or 633 (ATTO-TEC *AD 550-31, and* Sigma 01464-1MG-F, respectively). The oligonucleotides were conjugated with the esterified fluorophores using terminal deoxynucleotidyl transferase to produce labeled probes in an overnight reaction at 37 °C. Probes were then purified by precipitation and resuspended in a 15 μl final volume.

### smFISH protocol
A protocol for smFISH in the testis was kindly provided by the Yamashita lab, and adapted accordingly. Testes were dissected as described before and fixed in 4% FA/1x PBS for 30 min. Afterwards, the samples were washed three times 10 min in 1x PBS + 0.2%Tween 20 (PBT) and

rinsed in Wash Buffer A (10% Formamide, 2x SSC, 0.2% Tween 20). The samples are then transferred to hybridization buffer (2X SSC, 10% dextran sulfate, 1 mg/mL tRNA, 2 mM vanadyl ribonucleoside-complex (NEB, S142), 0.5% BSA, 10% formamide). Oligo probes were added in a concentration of 1:125 and hybridized overnight at 37 °C, 800 rpm. After hybridization, the samples were washed once for 15 min in Wash Buffer A, once for 15 min in 2x SSC, and once for 15 min in 1x PBT. For additional marker detection, an antibody staining was performed after the smFISH. Finally, the samples were mounted in Vectashield mounting media (Vector Laboratories, H1000).

## smFISH image processing

Acquired image stacks were processed using the RS-FISH plugin (v2.3.5) in Fiji (v1.54i). Stacks were processed in 3D, with an anisotropy coefficient of 1.00, and RANSAC fitting. Sigma = 1.5, support region radius = 3, inlier ratio = 0.1, max error = 1.50. DoG threshold was adjusted on a per-stack basis, as background signal varied per condition. Additionally, detections outside the testis tissue were removed for clearness, including those detected from high autofluorescent signals in trachea, lipid droplets, and laser backscatter near the coverslip.

## in vivo validations

TFs were perturbed in the 7 day old adult testis following temperature shift-induced activation of the GAL4 system. For soma and germline specific CRISPR mutagenesis, virgin flies from ptc-GAL4,UAS-Cas9 and nos-GAL4::VP16,UAS-Cas9 respectively were crossed with U6:3-gRNA-e transgenic males[106]. For control experiments, the GAL4,UAS-Cas9 virgin flies were crossed to $w^{1118}$ males. The F1 progeny were reared at 18 °C to suppress GAL4 activity. Upon eclosion, the flies were shifted to 29 °C for 7 days to activate the GAL4-induced CRISPR mutagenesis, providing spatiotemporal control of the gene knockout. All flies including controls were age-matched and treated identically. To validate TF perturbations using independent genetic tools, RNAi-mediated TF knockdowns in the soma or germline were performed by crossing virgins of c587-GAL4;vasa-EGFP or nos-GAL4::VP16,vasa-EGFP with UAS-RNAi males respectively. For the control experiment, the GAL4 virgins were crossed to UAS-mCherry$^{RNAi}$ males. The resulting F1 progeny from RNAi crosses were temperature shifted and treated identically to the CRISPR regime. The efficiency of knockout was confirmed using smFISH and the resulting phenotypes were characterized by quantifying total germ cell number (upto stage-16-stage spermatogonia), number of GSCs and GSC diameter (number of pixels at the same magnification and resolution) in both TF perturbed testes and their respective controls. Similarly, RNAi mediated knockdown of Wnt components were carried out using upd-GAL4, c587-GAL4 and nos-GAL4::VP16 in hub, soma or germline respectively. The KDs were carried out in two regimes: (1) A temperature-shift regime identical to the TF RNAi structure (18 °C to 29 °C post-eclosion for 7 days) with a temperature-sensitive GAL80 repressor (tub-GAL80$^{ts}$) to suppress the GAL4 activity in upd-GAL4 and c587-GAL4 testes. (2) a constitutive GAL4 activity where crosses were maintained at 29 °C from embryogenesis to 7-day-old-adults driving the expression of UAS-RNAi constructs resulting in KDs. GAL80ts was not used for germline-specific knockdowns as the nos-GAL4::VP16 driver is insensitive to GAL80$^{ts}$. The resulting phenotypes were characterized by quantifying the number of GSCs and were profiled against their respective age-matched control.

## Quantification of smFISH signal

CRISPR-mediated TF knockdown efficiency was assessed by smFISH, with signal quantification performed using the RS-FISH plugin. smFISH dot counts were measured in individual GSCs or CySCs and compared to controls. For ovo and klu knockouts, stg and nos transcripts were quantified per GSC. In the case of ci RNAi, wg and DWnt6 smFISH signals were quantified across the entire imaged testis section.

## Germline quantifications

Only testes exhibiting smFISH based knockouts in GSCs or CySCs were included for downstream germline quantifications. Immunostaining was performed alongside smFISH using Fas3 and Vasa antibodies to label the hub and germline, respectively. Total number of germ cells up to the stage-16 spermatogonia was manually quantified based on the Vasa staining to assess potential germline depletion in TF perturbations relative to the controls. In addition to quantifying the total germ cells, the number of GSCs in physical contact with the hub was manually counted and their diameters at the largest visible cross-section were measured using the measurement tool in Fiji to ensure consistent comparison between the TF perturbations and their respective controls.

## Generation of transgenic fly lines

The DWnt6 R6 + R7 regulatory region (dm6: 2 L:7330291-7331241) was amplified using the NEB Q5 polymerase and cloned into the pENTR-TOPO by directional cloning (Invitrogen Cat. No. K2400-20). Positive clones were identified by EcoRV control digests. Positive vectors were cloned in an LR clonase-based Gateway cloning strategy (Invitrogen Cat. No. 11791-020) into the pBPGUw_mCherry. Positive colonies were selected by EcoRI and KpnI digest, verified plasmids were injected by BestGene to generate transgenic lines.

## Immunohistochemistry

Testes were dissected from 7 day old flies in 1xPBS and fixed in 4% formaldehyde at RT for 20 min. Testes were then washed 3x10min in 1x PBS + 1% Triton X-100 (PBT), blocked with 1x PBS + 5% BSA for 30 min. Primary incubations were performed at 4 °C overnight. Primary antibodies were washed off 3×10 min in PBT. Testes were then incubated with secondary antibodies at room temperature for 2 h. Next, testes were washed 3x10min with PBT and mounted in Vectashield (Vector Laboratories, H1000).

## Antibodies

The following primary antibodies used in this study: Rabbit anti-Vasa (1:200, sc-30210, Santa Cruz), Mouse anti-Fas3 (1:100, 7G10, DSHB), Rabbit anti-GFP (1:500, A11122, Thermo Fisher Scientific), Chicken anti-GFP (1:300, 600-901-215, Rockland), Mouse anti-Ecad (1:300, 5D3, DSHB), Mouse anti-Eya (1:100, eya10H6, DSHB), Rabbit anti-β-Galactosidase (1:1000, Cappel labs). The following antibodies were kindly provided: Guinea Pig anti-Tj (1:10000) by Dorothea Godt, Rat anti-Org-1 (1:100) by Manfred Frasch. Alternatively, all primary antibodies used can be found as Supplementary Table 2.

The secondary antibodies used in this study: Donkey anti-Rabbit conjugated with Alexa 488, Goat anti-Rat conjugated with Alexa 594 (3:500, Jackson ImmunoResearch, PA), Goat anti-Mouse conjugated with Alexa 568, Goat anti-Guinea Pig conjugated with Alexa 647 (1:250, Thermo Fisher Scientific), Rhodamine Phalloidin (1:500, R415, Thermo Fisher Scientific), DAPI (1:500, Thermo Fisher Scientific). Immunofluorescent images were captured on a Leica TCS SP8.

## Statistics and reproducibility

Based on Exp1 and Exp2, sequencing depth was fitted to Michaelis-Menten kinetics and appropriate sequencing flow cells were chosen to reach 90% sequencing saturation for Exp3 and Exp4. GSC quantifications were compared to controls by two-sided independent $t$-tests using the statannotations package (v0.7.1). All in vivo cell quantifications were performed on $n(\geq)5$ biological replicates per perturbation with each replicate corresponding to a single testis. Immunofluorescence stainings and smFISH were performed at least twice, and reproducible in each case. Images were only linearly increased in brightness to enhance visibility of markers. Furthermore, no statistical method was used to predetermine sample size, and no data were excluded from the analyses. Experimental samples were all included or

randomly picked. The investigators were not blinded to allocation during experiments and outcome assessment.

## Reporting summary

Further information on research design is available in the Nature Portfolio Reporting Summary linked to this article.

## Data availability

The 10x Genomics Multiome sequencing raw data and processed files generated in this study have been deposited in the GEO database under accession number GSE277132. The quantifications on in vivo experiments in this study are provided in the Source Data file provided with this paper. Transcriptome, chromatin accessibility, eRegulons, and further analyses can be explored on our *Drosophila* testis Atlas Web Application (dAWA) at https://dawa.cos.uni-heidelberg.de. Source data are provided with this paper.

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

## Acknowledgements

We thank Michael Boutros for sharing the CRISPR fly lines. Further we thank the Bloomington Stock Center for fly lines and the DSHB for antibodies. We thank Dorothea Godt for the Tj antibody, Manfred Frasch for the Org-1 antibody. We would like to thank the CellNetworks Deep Sequencing Facility of Heidelberg University, in particular David Ibberson, for invaluable help with the Multiome experiment, library preparation and sequencing, and GeneCore at EMBL Heidelberg and the NGS at DKFZ for sequencing. The authors acknowledge support by the state of Baden-Württemberg through bwHPC. We would like to thank Jessica Velten for initial discussion on single cell RNA sequencing, Nicoletta Bobola, Siamak Redhai, Lauren M. Saunders and Britta Velten for critically reading the manuscript. This project was supported by the Deutsche Forschungsgemeinschaft (DFG SFB 873/B02, SFB 1324/A08).

## Author contributions

Conceptualization: P.v.N.S. and I.L. Experimental design: P.v.N.S., K.Y., J.B., F.P., M.B., and I.L. Experimental procedures: P.v.N.S., K.Y., L.Z.K., P.S.S., S.R.S., D.I., P.K., R.M.G., Q.W., N.E., V.M.B., K.D., X.G., S.P., and Aa.S. Data analysis: P.v.N.S., T.L., and P.S.S. Computational analysis: P.v.N.S., T.L., and An.S. Website development: T.L. and K.D. Writing: P.v.N.S., T.L., and I.L. Funding acquisition: I.L.

## Funding

## Competing interests

The authors declare no competing interests.
