## [Transparent Peer Review file · Nature Communications]

Dissecting the enhancer gene regulatory network in early *Drosophila* spermatogenesis

Corresponding Author: Professor Ingrid Lohmann

Version 0:

Reviewer comments:

Reviewer #1

(Remarks to the Author)

This work identified and characterised enhancer Regulons (eRegulons) involved in transcriptional regulation and chromatin architecture in the developing testis of *Drosophila*. Using integrated single-nucleus transcriptome and chromatin accessibility profiling of ~10,000 nuclei, the authors identified 138 high-quality, cell-specific eRegulons in germline and somatic cells. They functionally validated the transcription factors central to 14 networks and highlighted extensive co-regulation of genes via shared regulatory regions. They revealed that the Wnt signaling pathway, mediated by the TF Pangolin/Tcf, regulates germline stem cell numbers. To support public accessibility for their data, they developed the dAWA web tool, facilitating analysis of chromatin, transcriptomic, and regulatory interactions. Overall, I found the work would be useful for scientists who are interested in *Drosophila* testis biology.

Major comments:

1. The paper is very long—it has 67 pages of single-spaced main text and a total of nearly 20,000 words, without labelling line numbers. While I appreciate the detailed descriptions of multiple transcription factors (TFs) and eRegulons, the main conclusions are somewhat diffuse, making it difficult to identify the central points of the manuscript without repeatedly referring to the abstract and the last paragraph of introduction.
2. Figure 1b: There are a large number of cells that do not map to the Fly Cell Atlas testis data. While this is understandable for single-cell data, could you please hypothesize why this might be the case? A related question is: how did single-nuclei sequencing eliminate spermatids? Was this intentional, or is it simply a technical bias? Results are results but it is a bit strange that no spermatids were profiled given some are located near spermatocytes. It also appears that only pre-meiotic cells are profiled. If this is the case, the paper needs to clearly state this, in addition to the title, as spermatids and sperm are integral parts of spermatogenesis.

Specific comments:

1. “13,034 protein-coding genes, 5,839 non-coding RNAs and 46,619 accessible genomic regions” -this seems like a lot. How many of these are spurious?
2. Figure 1D and 1E: How is the number of “unique” genes in each cluster defined? It is unclear what “unique” means in this context.
3. “has predominantly a low enrichment of transcriptional start sites (TSS) (<1.5) (Supplementary Fig. 2b-d), which is generally a metric for the signal-to-noise ratio in snATAC-seq profiles⁴⁶. However, extensive remodeling and compaction of chromatin occurs in the germline”. The argument is confusing. The authors did not generate data from spermatids and sperm, where drastic chromatin remodeling and compaction occur. How much remodeling and compaction take place during the early stages of germ cell development?
4. “In this way, we identified 46,619 accessible chromatin regions of 500 bp in length in both the germline and somatic differentiation (Fig. 2c),” – Are these 46,619 accessible chromatin regions non-overlapping? The reason for asking is that peak discovery can overestimate peaks if a region is counted multiple times. If not, great, but it also sounds to be a lot of open chromatin regions.

5. "The number of accessible regions in the germline peaks at stage 16 spermatogonia (Sp-16) " – this is a very interesting observation. Does gene expression also peak at stage 16 spermatogonia, or does it occur at slightly later stages? At the whole-genome level, what is the relationship between open chromatin levels and gene expression levels?
6. "Finally, we found that most genes are associated with multiple accessible regions (Fig. 2d)," – how was this association calculated?
7. "In summary, our integrated snATAC- and snRNA-seq approach transforms our understanding of gene regulation by revealing the dynamic chromatin landscape" – this seems to be a bit of an overstatement.
8. "In primary (1°Scs) and secondary spermatocytes (2°Scs), a broader range of eRegulons is active." – the authors only explain what 1°Scs and 2°Scs mean on page 12. This should be mentioned earlier, as this nomenclature is used starting from Figure 1.
9. There are many other instances where abbreviations are either explained later, explained multiple times, or not explained at all in the paper. For example, what are Gbs (or GBs in some texts), Sp-2, or Sp-4? I had to guess their meanings in many sections of the manuscript.
10. Figure 5a is not readable. Therefore, it is unclear how Figure 5a "suggests a cooperative binding between various groups of TFs that bind a common set of target regions to jointly regulate a common set of genes in specific lineages." Given that eRegulons use relatively large regions for prediction, shared profiles do not necessarily indicate the use of a common set of target regions.
11. In the first paragraph of the Discussion, the description of autonomous and non-autonomous gene regulation is somewhat confusing, and it is unclear what these terms refer to. While the authors briefly mentioned these concepts in the introduction, there is no mention of them in the results section.

Reviewer #2

(Remarks to the Author)

In this manuscript, Sanchez et al., perform detailed and insightful profiling and bioinformatic analysis of the transcriptome and chromatin accessibility from *Drosophila* testis tips. They use SCENIC+ to identify novel eRegulons in the germline, including for the transcription factor *Ovo* which has been shown to be essential for female germline development. They further go on to use LIANA+ to identify cell-cell signaling pathways, identifying the Wnt signaling pathway as playing a role in germline stem cell maintenance through hub cell signaling. The bioinformatic analysis is rigorous and of use to the larger scientific community, providing significant insights. The authors mention that they have created a web application to facilitate the use of the data, which is particularly helpful and will be appreciated by the community, however that website is not currently functional and thus cannot be evaluated. While this atlas provides a high-quality, resource for the community, additional biological validation is needed to support some of the major claims within the manuscript. Specifically, *in vivo* data based on RNAi/CRISPR perturbation needs additional controls, and phenotypic analysis, and explanation to be convincing. Overall, this manuscript is well written, of significant interest to the community, but additional experiments are necessary to support some of the biological claims as they are currently written.

Major Points:

1. Additional methodological details for CRISPR experiments are necessary, including crossing schemes and generation details. Were all CRISPR experiments performed at varying temperatures (as shown in Fig. 4b and 5f)? For each CRISPR experiment, was the F1 or F2 generation used, and how were knockouts validated? If the F1 generation is used (as is indicated by Fig. 4b), knockouts will be heterogenous, and thus should be validated for each experiment (see point 2). For all CRISPR and RNAi experiments, was the driver crossed in through the male or virgin female? Particularly for nanos driven constructs, maternal deposition of the GAL4 results in significantly different (and earlier) embryonic phenotypes than if the driver was carried through the male.
2. Validation of each CRISPR knockout line is necessary, especially when the RNAi and CRISPR phenotypes are not consistent with each other. At a minimum, staining using smFISH, HCR, FISH or other method of the targeted gene should be done to show a loss of transcript in the effected tissue. This validation should be done for each experiment, particularly if stable knockout lines are not maintained. Ideally each mutant and WT tissue that is used for quantification should be stained with the targeted gene to show loss (or presence) of transcript. Quantification of the knockdown (or knockout) efficiency, using qPCR, staining of the target transcript, or other method should be added for each line.
3. mCherry RNAi is not an appropriate control for CRISPR experiments, and instead these experiments should be repeated with appropriate matched controls (either age matched lines carrying Cas9/mCherry-gRNA and the driver, or age-matched sibling controls with Cas9/target-gRNA and no driver).
4. A more detailed description of phenotypes is necessary- "thinner" and thicker (or increased/reduced testes size) are not very satisfying when describing the biological impact of the knockouts. Information on germ cell number (are testes thinner because they have less germline?) or an increase in cell size (are testes thicker because they have larger cells or more germline?) is needed. "Thinner" testes are common in older males, and thus age-matched controls and germ cell counts for these experiments are essential. Similarly, in supp. Fig 5j', is the enlarged tip due to an increase in GSCs, CySCs, GB, cysts, or spermatocytes?
5. In figure 5g, the control was kept at 29 degrees but both the *ovo* and *klu* CRISPR mutants were kept at 18 degrees, then presumably moved to 29 degrees in adulthood (based on figure labels). WT tissue at 29 degrees is not an appropriate

control for CRISPR mutants at 18 degrees. Control lines should be subjected to the same temperatures, at the same times in development, as the mutant lines for all experiments. Additionally, aged-matched sibling controls for both Fig. 5f and 5g would be more appropriate than vasa-EGFP lines (see point 3).

6. More in vivo evidence is needed to support the claim that “ovo and klu very likely contribute to GSC maintenance in the testis”, such as quantitation of GSC divisions, quantitative loss of the germline, or other such biological phenotypes in addition to the reduced testis size in these mutants. While the loss of stg and cher transcript is interesting, this alone does not suggest a role in GSC maintenance.

Minor Points:

1. In Figure 1b, it would be helpful to label all cell types in the UMAP (such as the spermatids, muscle, and epithelial cells) to show the enrichment of early testes-specific cell populations compared to the FCA dataset.
2. Showing staining for both ovo and bam transcript in the same tissue would be helpful in confirming that ovo expression peaks in GSCs and is lost in cells with increasing amounts of bam (Fig. 1f,g) as stated in text.
3. In figure 2a, label the mixed germline/soma cluster on the UMAP, same as it is labeled in supp. Fig. 2a.
4. Figure 2d is incorrectly cited in the sentence “Interestingly, chromatin accessibility increases in the somatic lineage accompanying the 2°Sc stage (Fig. 2d), a trend that is also observed at the transcriptome level (Fig. 1d, e).” Fig. 2C seems to be the correct figure citation for this sentence?
5. The sentence “This suggests a shift in germline dependency to the soma, consistent with an increase in metabolic gene expression (Supplementary Fig. 1f).” may be overstated without additional in vivo evidence.
6. Table 1 is missing
7. The gRNA sequence for Klu or pan is not written in table S1

Reviewer #3

(Remarks to the Author)

In this manuscript by van Nierop y Sanchez et al., they combined single-nucleus sequencing and single cell ATAC seq to infer enhancer regulatory network in the Drosophila testis (both somatic cells and early germ cells). This provides a rich resource for researchers in the field to investigate gene regulation during cellular differentiation of this tissue. Through the combination of RNAseq and ATACseq, they identified the network of transcription factors and their targets in germline and somatic cells. This provides an impressively convincing analysis of how gene expression is regulated in this tissue during cellular differentiation.

The authors moved onto functional validation of their finding (Fig 4). Here, however, the phenotypes are not reported in a manner that allows reproducibility. The graphs (f, j, n etc) categorize the phenotype as ‘major phenotype’ vs. ‘minor phenotype’, but it is very unclear what was exactly measured. The images provided in Fig4 does not seem to reveal clear phenotype, other than slightly thinner testes, which can be easily affected by other parameters, such as age, temperature etc. Accordingly, I am unsure what kind of phenotype was categorized as ‘major phenotype’. They should use more concrete, measurable criteria such as ‘testis width’ (although this can be influenced by how they mount the sample on the slide glass), GSC number, or number of germ cells etc. Such careful analysis would be particularly important, because their results/conclusions somewhat differ from previous studies (I agree that experimental setting, especially the timing of depletion, can lead to very different outcome, but without more precise measurement, the readers won’t even know how different the reported phenotypes are. Also I am surprised that Mad depletion did not cause complete germ cell loss.

Fig5 presents an interesting idea of multiple TFs co-regulating the targets. However, again, validation requires more rigorous experimentation. Immunostaining is far from quantitative, especially when comparing the staining between tissues. Thus it cannot be used to detect downregulation in the mutant. Single molecule RNA FISH to count the number of transcripts will be more accurate.

Finding described in Fig6 regarding the potential involvement of Wnt is interesting. But I cannot see fz-GFP in the germline, as the authors describes (Fig6d). Please clarify which signal was regarded as the germline-FzGFP. Similarly, it is unclear Otk and Arr signals that the authors claim to be in GSCs are actually above the background level. Although the authors state that Fz is not expressed in GSCs at an RNA level but the protein is in GSCs, they use germline-RNAi to remove Fz from GSCs. This is very confusing as it is unclear how one can knockdown RNAi that is not present in that cell type. While the expression of Wnt pathway genes in the hub cells is very interesting, the functional study remains inconclusive.

In Fig7, I don’t understand why Pan RNAi causes such a drastic complete GSC loss (Fig7), whereas depletion of its upstream regulators in the Wnt pathway does not lead to GSC loss.

In sum, this paper presents important analysis of transcription network in the Drosophila testis, with a tour de force approach using combined snRNA seq and snATAC seq. These analyses may already be worth publishing this paper, so I am disappointed that the following analyses are done rather poorly. I hope that the authors solidify functional validations. If such analysis turns out to be too extensive, I’d rather want to see a fewer stories but done carefully. For example, they can cut the Wnt stories but the earlier part should be done more rigorously.

Version 1:

Reviewer comments:

Reviewer #1

(Remarks to the Author)

The authors did a very good job addressing my comments and concerns. The readability of the manuscript has also greatly improved. I have no further questions.

Reviewer #2

(Remarks to the Author)

In this revision of Sanchez et al., the authors have made substantial efforts to address all reviewer comments. They conducted additional experiments, quantitative analyses, and validations, which strengthen and clarify their scientific findings. The manuscript has also been significantly streamlined, with over ten pages of text removed, resulting in a clearer presentation, a more detailed methods section, and a stronger set of results. All major concerns raised by reviewers have been fully addressed, and we have no further concerns.

Reviewer #3

(Remarks to the Author)

This is a revised manuscript from the Lohmann group, describing their characterization of *Drosophila* male germ cells by snRNA seq and snATAC seq. This identified 'eRegulons', the distinct sets of enhancers that regulate gene expression in a cell type-specific manner.

This study provides an important resource for the field. They have improved their cytological analysis, and the manuscript appears to be in good shape overall. I do have a few more comments.

Specific comments

-Multiple studies have repeatedly demonstrated that ovo mutant does not have discernable phenotypes in male. Can you speculate/discuss on this possible discrepancy?

-They used 'GSC diameter' as a phenotypic criterion, but I don't believe GSC size is an established phenotype that indicates 'GSC dysfunction'. GSC diameter would be influenced if they are arrested/slowed down in the G1 phase of the cell cycle, but it is hard to imagine all the mutants they examined exhibit the same 'smaller GSC' phenotype. Also, based on the images provided, the GSC diameter does not seem to be smaller in mutants (as much as shown in the graph). Can you provide a better rationale why they chose GSC diameter, and what it (small diameter) may mean? Additionally, the unit of GSC diameter cannot be 'pixels', as is shown in the figures (Fig 4).

-Line 117: mention of 'secondary spermatocytes': do they really mean 'haploid germ cells after meiosis I'? (which is the definition of secondary spermatocytes). If so, the transition from spermatogonia to 'primary spermatocytes' is not described.

POINT-BY-POINT RESPONSE TO THE REVIEWERS:

We would like to thank all the reviewers for their constructive criticism and important suggestions, which have helped to improve the manuscript significantly. We have included new data and have changed many aspects according to reviewer suggestions, which we describe in detail below. I would like to highlight here, that we have substantially shortened the manuscript by 13 pages, which was one of the requests of reviewer #1. This improved the flow and clarity of the manuscript, however also resulted in a substantial re-writing of the text.

Reviewer #1: thought the study is well suited for Nature Communication, as it identified and characterised enhancer Regulons (eRegulons) involved in transcriptional regulation and chromatin architecture in the testis of *Drosophila*.

There were a few concerns raised by this reviewer, which we addressed in the following way:

Major Points:

1. The paper is very long—it has 67 pages of single-spaced main text and a total of nearly 20,000 words, without labelling line numbers. While I appreciate the detailed descriptions of multiple transcription factors (TFs) and eRegulons, the main conclusions are somewhat diffuse, making it difficult to identify the central points of the manuscript without repeatedly referring to the abstract and the last paragraph of introduction.

We thank the reviewer for his comment. We have now shortened the manuscript substantially, which led to a dramatic re-writing of the manuscript (reduction from 67 to 54 pages including Supplement). We also included line numbers for clarity. We think that the conclusions are now much clearer and the manuscript is much easier to follow.

2. Figure 1b: There are a large number of cells that do not map to the Fly Cell Atlas testis data. While this is understandable for single-cell data, could you please hypothesize why this might be the case? A related question is: how did single-nuclei sequencing eliminate spermatids? Was this intentional, or is it simply a technical bias? Results are results but it is a bit strange that no spermatids were profiled given some are located near spermatocytes. It also appears that only pre-meiotic cells are profiled. If this is the case, the paper needs to clearly state this, in addition to the title, as spermatids and sperm are integral parts of spermatogenesis.

We thank the reviewer for this comment and the opportunity to clarify. The nuclei shown in Figure 1b represent those passing the initial, relatively loose thresholds described in the Materials & Methods, and this panel is intended only as a general overview of the sequenced material. The large number of barcodes in the central region of the UMAP that do not overlap with the FCA testis dataset likely result from differences in background RNA content. While both datasets are derived from *w¹¹¹⁸* flies at day one post-eclosion, different nuclei isolation protocols were used. Our protocol, optimized for multiome analysis, required permeabilization conditions that allow Tn5 transposase access for ATAC-seq while minimizing RNA leakage—constraints not present in the FCA protocol.

We also note that the central cloud of barcodes corresponds to nuclei expressing spermatid-specific *cup* genes. Given that spermatids are transcriptionally less active than earlier germline stages, the associated background RNA may disproportionately influence their apparent identity, suggesting that this region of the UMAP may reflect a mixture of late spermatids and RNA-containing empty droplets. Regarding the reviewer's side question: we did detect spermatids in our initial dataset (see Materials & Methods snRNA-seq processing section lines 584-586, and Supplementary Table S3), but they were excluded from downstream analyses due to their low transcriptional activity and a discontinuity in the pseudotime trajectory between second spermatocytes and late spermatids. Additionally, most spermatids were physically removed during sample preparation using 40 μ m Flowmi filters, which likely

retain these cells as their flagella get trapped in the filter matrix. This depletion was confirmed by microscopy imaging of the cell suspension before and after filtration.

Lastly, we do include meiotic nuclei in our dataset (e.g., late spermatocytes, positive for markers *twc* and *bol*; see Figure 1g, and the onset expression of male fertility factors kl-2 and kl-5 in Figure 1d). However, our functional and perturbation analyses from Figure 4 onward focus specifically on early germline and somatic lineages at the testis apical tip, which may have contributed to the impression that later stages were not represented.

Specific comments:

1. *“13,034 protein-coding genes, 5,839 non-coding RNAs and 46,619 accessible genomic regions” -this seems like a lot. How many of these are spurious?*

We thank the reviewer for pointing this out. The 13,034 protein-coding genes and 5,839 non-coding RNAs we report are detected in at least three of the 10,335 nuclei, which does result in some genes being detected very sparsely. This observation is consistent with the Fly Cell Atlas, where the testis exhibits the highest number of expressed genes overall—over 11,000 with 10x and more than 12,000 with Smart-seq2—reflecting the inclusion of diverse cell types such as terminal epithelium, muscle, and spermatids. We agree that summarising gene expression across such heterogeneous cell types is not informative, given the large differences in transcriptional activity between, for example, spermatocytes and hub cells. In response, we have replaced the previous line plot with violin plots in Figures 1d–e, which more accurately represent the distribution of detected genes per nucleus across cell types. Regarding the 46,619 accessible genomic regions, we thank the reviewer for prompting clarification. This number reflects peaks called with MACS2 using default parameters ($q = 0.05$), as detailed in the Materials & Methods. This is comparable to the approach used in the FlyBrain study (doi: 10.1038/s41586-021-04262-z), which identified ~95,000 peaks. While the brain comprises numerous neuronal subtypes, the testis undergoes extensive chromatin reorganisation during spermatogenesis, which likely contributes to the high number of accessible regions observed.

2. *Figure 1D and 1E: How is the number of “unique” genes in each cluster defined? It is unclear what “unique” means in this context.*

Thank you for highlighting this, we changed “unique genes” to just “genes”, better representing what we wanted to show.

3. *“has predominantly a low enrichment of transcriptional start sites (TSS) (<1.5) (Supplementary Fig. 2b-d), which is generally a metric for the signal-to-noise ratio in snATAC-seq profiles⁴⁶. However, extensive remodeling and compaction of chromatin occurs in the germline”. The argument is confusing. The authors did not generate data from spermatids and sperm, where drastic chromatin remodeling and compaction occur. How much remodeling and compaction take place during the early stages of germ cell development?*

We appreciate the reviewer’s thoughtful comment. Indeed, chromatin becomes highly compacted during spermiogenesis due to histone replacement by protamines. However, early germline cells in the *Drosophila* testis, GSCs, gonialblasts, and early spermatogonia, maintain a euchromatic and transcriptionally active state. This is evidenced by the enrichment of active histone marks such as H3K4me2 at promoters in *bam* mutant testes, which are enriched for these early germline stages (Anderson et al., 2023, *eLife* doi.org/10.7554/eLife.89373.3.sa0). The modest TSS enrichment we observe is consistent with this chromatin configuration and reflects biologically meaningful accessibility rather than technical noise. For this reason and due to the extensive rewriting of the main text, we removed the technical focus on nuclei with a low TSS score, thereby also addressing this confusion.

4. *“In this way, we identified 46,619 accessible chromatin regions of 500 bp in length in both the germline and somatic differentiation (Fig. 2c),” – Are these 46,619 accessible chromatin regions non-overlapping? The reason for asking is that peak discovery can overestimate peaks if a region is counted multiple times. If not, great, but it also sounds to be a lot of open chromatin regions.*

Thank you for noticing this, and indeed, the direct output from MACS2 does give peaks of varying lengths that may overlap between cell type clusters, thereby overestimating the accessibility of chromatin. We would like to clarify that the 46,6519 genomic sub-regions are part of 29989 non-overlapping accessible peaks discovered by MACS2. That is, a single accessible peak can compose multiple non-overlapping accessible regions. We have now clarified that in the main text and write “*We defined 25 transcriptionally distinct clusters and identified 29989 non-overlapping accessible chromatin peaks using MACS2...*” (lines 163-164).

5. “*The number of accessible regions in the germline peaks at stage 16 spermatogonia (Sp-16) – this is a very interesting observation. Does gene expression also peak at stage 16 spermatogonia, or does it occur at slightly later stages? At the whole-genome level, what is the relationship between open chromatin levels and gene expression levels?*”

We would like to thank the reviewer for their shared interest in our findings. When chromatin accessibility reaches its highest point at stage-16 spermatogonia (Figure 2c, right inset), this is accompanied by the largest change in differentially expressed genes (Figure 1d, right lower inset, highest increase in DEGs and lowest DEGs compared to state-8 spermatogonia). However, the height of transcript abundance occurs later at the secondary spermatocyte stages, where it has been shown that transcripts are accumulated and used post-meiotically when chromatin is being remodeled and becomes less accessible for the transcription machinery.

Regarding the relationship between open chromatin levels and gene expression levels, we observe two dynamics in the cell types that we captured; In the soma, maybe more to what is expected, chromatin accessibility precedes transcription levels (Figure 2c and Figure 1e), in line with that chromatin has to become accessible for TFs and the transcriptional machinery in order to facilitate transcription. In the germline however, genome accessibility increases initially while the number of total genes expressed remains the same, although there is a shift in the exact genes (Figure 1d, right lower inset). Upon pre-meiotic entry, genome accessibility starts to decrease (Figure 2c) while the number of different genes increases up to the secondary spermatogonial stages (Figure 1d). This suggests extensive chromatin remodeling events occurring while the transcriptional machinery prepares all the necessary components before they become inaccessible.

6. “*Finally, we found that most genes are associated with multiple accessible regions (Fig. 2d),” – how was this association calculated*”

Thank you for bringing this up. We have now clarified in the main text that these associations are from the correlative analysis of SCENIC+ (lines 180-182) (explained in the following section in the main text), where accessible regions correlate with the transcript levels of neighboring genes ($\pm 50\text{kb}$ of gene TSS). Furthermore, we refer the readers to the Materials and Methods section for more details.

7. “*In summary, our integrated snATAC- and snRNA-seq approach transforms our understanding of gene regulation by revealing the dynamic chromatin landscape” – this seems to be a bit of an overstatement.*”

We agree with the reviewer that this was an over-statement. In the course of the re-writing of the manuscript, we have also changed this conclusion to: “*In summary, by linking TF motifs to accessible regions within single nuclei, we reveal how regulatory potential shifts over time, consistent with stage-specific transcriptional programs and lineage progression. These findings would not have been possible by investigating the transcriptomics alone.*” (lines 204-206)

8. “*In primary (1°Scs) and secondary spermatocytes (2°Scs), a broader range of eRegulons is active.” – the authors only explain what 1°Scs and 2°Scs mean on page 12. This should be mentioned earlier, as this nomenclature is used starting from Figure 1*”

We thank the reviewer for this comment. We now introduce the nomenclature already in the Introduction and keep the written out version and the abbreviations in the whole manuscript for clarity.

9. *There are many other instances where abbreviations are either explained later, explained multiple times, or not explained at all in the paper. For example, what are Gbs (or GBs in some texts), Sp-2, or Sp-4? I had to guess their meanings in many sections of the manuscript*

The same comment as above: We now introduce the nomenclature already in the Introduction and keep the written out version and the abbreviations in the whole manuscript for clarity.

10. Figure 5a is not readable. Therefore, it is unclear how Figure 5a “suggests a cooperative binding between various groups of TFs that bind a common set of target regions to jointly regulate a common set of genes in specific lineages.” Given that eRegulons use relatively large regions for prediction, shared profiles do not necessarily indicate the use of a common set of target regions.

We thank the reviewer for this comment and agree that the original Fig. 5a was unreadable. For readability, we have split the matrix in two sections. We now show in Fig. 5a the TF common target regions in the germline, while the new somatic matrix is moved to Supplementary Fig. 5d. In addition, we also agree that the original formulation was overstated. We have revised the text to remove the term "cooperative" and now describe the observation more cautiously, and say that “.. TFs may coordinate via shared cis-regulatory elements to drive lineage-specific expression programs.” (lines 293-294)

As an explanation: Regarding the size of regions for predictions (500bp), we understand the concern. However, reducing the size of the accessible genomic regions to for example 50bp for correlative analysis would not result in a reduced overlap of TF-accessible regions. Instead, the total number would only increase, as the smaller regions would still mostly correlate with the same set of target genes, and therefore the common set of target regions would remain similar. Additionally, reducing the size of putative accessible regions/peaks would also severely compromise the identification of binding sites, as TF binding motifs might be cut in half and therefore not identified. Therefore, a compromise of 500bp has been chosen for regulatory region characterization, in line with previous work done in the fly brain.

11. In the first paragraph of the Discussion, the description of autonomous and non-autonomous gene regulation is somewhat confusing, and it is unclear what these terms refer to. While the authors briefly mentioned these concepts in the introduction, there is no mention of them in the results section.

To make this more clear, we have now changed this section of the Discussion to:

“The adult Drosophila testis is a powerful model for dissecting both autonomous (cell-intrinsic) and non-autonomous (intercellular) gene regulation due to its spatially organized and continuously renewing stem cell niche.”

Reviewer #2: noted that the study offers a comprehensive and insightful analysis of transcriptomic profiles and chromatin accessibility in Drosophila testis tips, supported by robust bioinformatic approaches. The reviewer pointed out that the atlas provides a high-quality, resource for the community. However, this reviewer also thought that additional biological validation is needed to support some of the major claims within the manuscript.

The concerns raised by this reviewer were addressed in the following way:

Major Points:

1. Additional methodological details for CRISPR experiments are necessary, including crossing schemes and generation details. Were all CRISPR experiments performed at varying temperatures (as shown in Fig. 4b and 5f)? For each CRISPR experiment, was the F1 or F2 generation used, and how were knockouts validated? If the F1 generation is used (as is indicated by Fig. 4b), knockouts will be heterogenous, and thus should be validated for each experiment (see point 2). For all CRISPR and RNAi experiments, was the driver crossed in through the male or virgin female? Particularly for nanos driven constructs, maternal deposition of the GAL4 results in significantly different (and earlier) embryonic phenotypes than if the driver was carried through the male.

We thank the reviewer for the detailed questions and appreciate the opportunity to clarify our experimental approach. To validate the efficiency of our CRISPR knockouts, we repeated all experiments and performed smFISH to confirm reproducible gene-specific signal reduction. Only lines with robust smFISH signals for the targeted gene were used in the final analysis; accordingly, foxo, pnt, and bowl were excluded due to unreliable probe performance. For all remaining genes, CRISPR knockouts were induced exclusively during adulthood by maintaining flies at 18 °C until eclosion and

subsequently shifting males to 29 °C. This strategy aligns with the adult-specific single-nucleus transcriptomic profiling, where the relevant TFs and eRegulons are active. Testes were dissected and analyzed seven days post-temperature shift, as in previous experiments. All data derive from the F1 generation; as illustrated in Fig. 4a, females expressing both GAL4 and UAS-Cas9 were crossed to males carrying ubiquitously expressed guide RNAs (U6 promoter). Full experimental details are now provided in the Materials and Methods section. (section is completely re-written; lines 242-285)

2. Validation of each CRISPR knockout line is necessary, especially when the RNAi and CRISPR phenotypes are not consistent with each other. At a minimum, staining using smFISH, HCR, FISH or other method of the targeted gene should be done to show a loss of transcript in the effected tissue. This validation should be done for each experiment, particularly if stable knockout lines are not maintained. Ideally each mutant and WT tissue that is used for quantification should be stained with the targeted gene to show loss (or presence) of transcript. Quantification of the knockdown (or knockout) efficiency, using qPCR, staining of the target transcript, or other method should be added for each line.

We thank the reviewer for this valuable suggestion. In response, we have conducted smFISH analysis across all targeted TF knockout genotypes and now provide a detailed description of the quantification procedure in the Materials & Methods section. To maintain consistency with our single-nucleus profiling of adult testes, all quantifications and phenotypic analyses were carried out exclusively in temperature-shifted animals, thereby isolating adult-specific effects. All CRISPR knockouts exhibit a highly significant reduction in the expression of the targeted TFs in 7-day-old testes, as demonstrated by the smFISH images and quantifications in Figure 4 and Supplementary Fig. 3. The *foxo*, *pnt* and *bowl* knockouts were excluded due to unreliable smFISH probe performance. Accordingly, phenotypes are consistently observed for all TFs with reproducibly reduced expression. While we also observe phenotypes in other TF knockouts and believe these are gene-specific, we have excluded them from the current manuscript due to insufficient expression validation. (section is completely re-written; lines 242-285)

3. mCherry RNAi is not an appropriate control for CRISPR experiments, and instead these experiments should be repeated with appropriate matched controls (either age matched lines carrying Cas9/mCherry-gRNA and the driver, or age-matched sibling controls with Cas9/target-gRNA and no driver).

We thank the reviewer again for this comment. To address the concern, we have now included the appropriate controls: since our GAL4 lines already carry Cas9, we crossed each GAL4; UAS-Cas9 line to *w¹¹¹⁸* and used the resulting age matched progenies for both quantification and phenotypic analyses.

4. A more detailed description of phenotypes is necessary- "thinner" and thicker (or increased/reduced testes size) are not very satisfying when describing the biological impact of the knockouts. Information on germ cell number (are testes thinner because they have less germline?) or an increase in cell size (are testes thicker because they have larger cells or more germline?) is needed. "Thinner" testes are common in older males, and thus age-matched controls and germ cell counts for these experiments are essential. Similarly, in supp. Fig 5j', is the enlarged tip due to an increase in GSCs, CySCs, GB, cysts, or spermatocytes?

We again thank the reviewer for these helpful suggestions. To clarify, all analyses were performed using age-matched controls and experimental animals, we precisely explain this now in Materials and Methods and show it in Figure 4a. In response to the reviewer's request, we have now quantified GSC numbers and total germ cell numbers up to the spermatocyte stage in both control and TF CRISPR knockout conditions. We also measured the diameters of GSCs under control and perturbation conditions. All analyses were conducted exclusively in temperature-shifted animals to focus on adult-specific effects. The results are presented in Fig. 4 and Supplementary Fig. 3. As we did not get the *pnt*, *foxo* and *bowl* smFISH probes to work, we excluded these from the Supplementary Figure 4, thus we also did not quantify germ cell number nor diameter. (section is completely re-written; lines 242-285)

5. In figure 5g, the control was kept at 29 degrees but both the *ovo* and *klu* CRISPR mutants were kept at 18 degrees, then presumably moved to 29 degrees in adulthood (based on figure labels). WT tissue at 29 degrees is not an appropriate control for CRISPR mutants at 18 degrees. Control lines should be subjected to the same temperatures, at the same times in development, as the mutant lines for all experiments. Additionally, aged-matched sibling controls for both Fig. 5f and 5g would be more appropriate than *vasa-EGFP* lines (see point 3).

We thank the reviewer again for their accurate and constructive remarks. In response, we have repeated the experiments using the appropriate controls and methodology. First, we excluded the Cheerio experiment previously shown in Fig. 5f, as Reviewer 3 correctly noted that antibody staining is not suitable for assessing target gene regulation. We have replaced this with smFISH analysis of *nanos* mRNA, a common downstream target of both *ovo* and *klu*. For both *nanos* and *stg*, we used the correct control genotype (*nos-GAL4; UAS-Cas9* crossed to *w¹¹¹⁸*) and performed all analyses on age-matched testes, raised at 18°C until adulthood and shifted to 29°C after eclosion to assess adult-specific effects (Fig. 5f, g). The quantification of smFISH signal is described in the Materials & Methods, and in both cases, we observed a highly significant reduction in mRNA levels of *nanos* and *stg*.

6. More *in vivo* evidence is needed to support the claim that “*ovo* and *klu* very likely contribute to GSC maintenance in the testis”, such as quantitation of GSC divisions, quantitative loss of the germline, or other such biological phenotypes in addition to the reduced testis size in these mutants. While the loss of *stg* and *cher* transcript is interesting, this alone does not suggest a role in GSC maintenance.

To provide evidence that *ovo* and *klu* are involved in GSC maintenance, we performed the following analyses: (1) We carried out CRISPR-mediated knockouts restricted to adulthood and quantified both GSCs and total germ cells up to the spermatocyte stage, using properly age-matched controls. This revealed a reduction in both GSC number and overall germ cell content. (2) We validated the efficiency of the knockouts by performing smFISH for *ovo* and *klu* in germline-specific knockout testes, confirming reduced expression of the targeted genes. (3) We assessed the expression of established GSC regulators (*nanos* and *stg*), which are predicted to be targets of *ovo* and *klu*, using smFISH, both of which showed significantly reduced expression following *ovo* or *klu* knockout. Together, these findings strongly suggest a role for *ovo* and *klu* in GSC maintenance via regulation of two genes known to be important for GSC maintenance. In response to the reviewer’s concern, we have moderated our conclusions to state that *ovo* and *klu* are likely to contribute to GSC maintenance. (new data are described in lines 314-319)

Minor Points:

1. In Figure 1b, it would be helpful to label all cell types in the UMAP (such as the spermatids, muscle, and epithelial cells) to show the enrichment of early testes-specific cell populations compared to the FCA dataset.

Thank you for pointing this out. We have now labeled the different cell clusters by merging annotations from the Fly Cell Atlas, and added our UMAP as a gray overlay.

2. Showing staining for both *ovo* and *bam* transcript in the same tissue would be helpful in confirming that *ovo* expression peaks in GSCs and is lost in cells with increasing amounts of *bam* (Fig. 1f,g) as stated in text.

We attempted smFISH for *ovo* and *bam* in the same testis, but the results were suboptimal. We also used *bam-GFP* fly lines; however, these lines exhibited a phenotype on their own, making them unreliable for our analysis. Nevertheless, we include an image of a *bam-GFP* testis stained with *ovo* smFISH (pink), GFP labelling *bam* positive cells and Fas3 (white) labelling the hub, showing that *ovo* transcripts accumulate in GSCs and Gbs and fade away in the *bam* positive cells..

3. In figure 2a, label the mixed germline/soma cluster on the UMAP, same as it is labeled in supp. Fig. 2a.

Thank you, the labeling has now been updated.

4. Figure 2d is incorrectly cited in the sentence “Interestingly, chromatin accessibility increases in the somatic lineage accompanying the 2°Sc stage (Fig. 2d), a trend that is also observed at the transcriptome level (Fig. 1d, e).” Fig. 2C seems to be the correct figure citation for this sentence?

We thank the reviewer for spotting this, we have now put in the correct figure citation.

While genomic accessibility can be read from both 2c or 2d, we overlaid the pseudotimes of the germline and somatic lineages in figure 2d to facilitate interpretation. However, as the somatic lineage has been binned into less time points, we refrained from putting exact cluster labels on the x-axis due to readability concerns, and instead show the latent time, which is scaled between 0 and 1. However, to accommodate interpretation off main transition points, we have now added the “spermatogonia”, “1° Sc”, and “2° Sc” labels at the top off the figure 2d.

5. The sentence “This suggests a shift in germline dependency to the soma, consistent with an increase in metabolic gene expression (Supplementary Fig. 1f).” may be overstated without additional *in vivo* evidence.

We toned down the statement, which now reads like that “This may indicate a shift in regulatory interplay between germline and soma, coinciding with increased expression of metabolic genes”

6. Table 1 is missing

We added the missing table.

7. The gRNA sequence for *Klu* or *pan* is not written in table S1

We added the missing information in table S1.

Reviewer #3: emphasized that this study provides a rich resource for researchers in the field to investigate gene regulation during cellular differentiation of the *Drosophila* testis. The reviewer pointed out that this is an impressively convincing analysis of how gene expression is regulated in this tissue during cellular differentiation.

This reviewer also had some concerns, which we addressed in the following manner:

1. The authors moved onto functional validation of their finding (Fig 4). Here, however, the phenotypes are not reported in a manner that allows reproducibility. The graphs (f, j, n etc) categorize the phenotype

as 'major phenotype' vs. 'minor phenotype', but it is very unclear what was exactly measured. The images provided in Fig4 does not seem to reveal clear phenotype, other than slightly thinner testes, which can be easily affected by other parameters, such as age, temperature etc. Accordingly, I am unsure what kind of phenotype was categorized as 'major phenotype'. They should use more concrete, measurable criteria such as 'testis width' (although this can be influenced by how they mount the sample on the slide glass), GSC number, or number of germ cells etc.

The same concern was raised by reviewer 2 (see above). To also explain here: all analyses were performed using age-matched controls and experimental animals, we precisely explain this now in Materials and Methods. As a more specific measurement of phenotypes, we have now quantified GSC numbers and total germ cell numbers up to the spermatocyte stage in both control and TF CRISPR knockout conditions. We also measured the diameters of GSCs and spermatogonia under control and perturbation conditions. All analyses were conducted exclusively in temperature-shifted animals to focus on adult-specific effects. The results are presented in Fig. 4 and Supplementary Fig. 4. For phenotypic quantification, we selected only testes that exhibited the expected phenotype based on smFISH-confirmed knockout efficiency of the targeted genes, ensuring that we analysed only affected tissues. Using this approach, we found that CRISPR mutagenesis of *stat* in the somatic lineage consistently resulted in smaller testes due to a reduced number of germ cells - a phenotype consistent with published data, but not with our initial observations. To verify this discrepancy, we repeated the experiment and again observed germ cell reduction, confirming that CRISPR mediated mutagenesis with *Stat92E* in the soma results in the same phenotype as in the study by Leatherman & DiNardo, 2010. We excluded now *pnt*, *foxo* and *bow1* from the manuscript, as the smFISH probe for these genes did not work, thus we could not prove reduced/absent expression of these genes. Thus we also did not quantify germ cell number nor diameter for these genes. (section is completely re-written; lines 242-285)

2. Such careful analysis would be particularly important, because their results/conclusions somewhat differ from previous studies (I agree that experimental setting, especially the timing of depletion, can lead to very different outcome, but without more precise measurement, the readers won't even know how different the reported phenotypes are. Also I am surprised that *Mad* depletion did not cause complete germ cell loss.

We agree with the reviewer, we have re-done the analysis, and present the results in Fig. 4 and Supplementary Fig.4. As shown by smFISH, the CRISPR targeted genes are strongly reduced in their expression, but expression is not completely absent. This is likely due to mosaicism from incomplete CRISPR efficiency, which has been reported before (Port et al, 2014). We now have re-written this part in the following way: "Although previous *Mad* mutant studies showed complete germline loss²⁷, the phenotype here was milder, likely due to mosaicism from incomplete CRISPR efficiency. Residual *Mad* expression in germ cells and preserved expression in the soma (Fig. 4b, c) support this interpretation and confirm lineage specificity of the system." (section is completely re-written; lines 242-285)

3. Fig5 presents an interesting idea of multiple TFs co-regulating the targets. However, again, validation requires more rigorous experimentation. Immunostaining is far from quantitative, especially when comparing the staining between tissues. Thus it cannot be used to detect downregulation in the mutant. Single molecule RNA FISH to count the number of transcripts will be more accurate.

We thank the reviewer, the comment is very similar to one of the comments of Reviewer 2. First, we excluded the Cheerio experiment previously shown in Fig. 5f, as Reviewer 3 correctly noted that antibody staining is not suitable for assessing target gene regulation. We have replaced this with smFISH analysis of *nanos* mRNA, a common downstream target of both *ovo* and *klu*. For both *nanos* and *stg*, we used the correct control genotype (*nos-GAL4;Cas9* crossed to *w¹¹¹⁸*) and performed all analyses on age-matched testes, raised at 18°C until adulthood and shifted to 29°C after eclosion to assess adult-specific effects (Fig. 5f, g). The quantification of smFISH signal is described in the Materials & Methods, and in both cases, we observed a highly significant reduction in mRNA levels of *nanos* and *stg*. (new data are described in lines 314-319)

4. Finding described in Fig6 regarding the potential involvement of Wnt is interesting. But I cannot see fz-GFP in the germline, as the authors describes (Fig6d). Please clarify which signal was regarded as the germline-FzGFP. Similarly, it is unclear Otk and Arr signals that the authors claim to be in GSCs are actually above the background level.

We thank the reviewer for this helpful comment. To avoid confusion, we have removed the antibody stainings and now show more details in the smFISH data demonstrating that the corresponding genes are lowly but specifically expressed in GSCs; importantly, we also show that this low-level expression is sufficient to elicit mild but consistent phenotypes upon RNAi knockdown, supporting a functional role in these cells.

5. Although the authors state that Fz is not expressed in GSCs at an RNA level but the protein is in GSCs, they use germline-RNAi to remove Fz from GSCs. This is very confusing as it is unclear how one can knockdown RNAi that is not present in that cell type. While the expression of Wnt pathway genes in the hub cells is very interesting, the functional study remains inconclusive.

We appreciate the reviewer's concern and acknowledge the apparent discrepancy between smFISH results and RNAi-based phenotypes. However, the absence of a detectable transcript by smFISH does not necessarily indicate that the transcript is absent or that RNAi is non-functional in that cell type. smFISH, while highly sensitive, has a defined detection threshold and may not reliably detect low-abundance or rapidly turned-over transcripts, particularly in small or transcriptionally dynamic cells like GSCs. By contrast, RNAi acts catalytically and can effectively target even minimal amounts of mRNA that fall below the detection limit of smFISH.

This mechanistic dissociation is supported by single-molecule imaging studies showing that mRNA degradation can occur even when transcripts are rare and spatially dispersed. Notably, Horvathova et al. (2017; Molecular Cell 68:615–625.e9) visualized RISC-mediated cleavage of individual mRNAs in real time and demonstrated that mRNA turnover can proceed in the absence of detectable transcript accumulation by conventional imaging methods. Thus, RNAi-based depletion remains a valid functional approach, even in cases where smFISH does not reveal the target transcript.

We have now adjusted the sections accordingly and included the respective reference.

5. In Fig7, I don't understand why Pan RNAi causes such a drastic complete GSC loss (Fig7), whereas depletion of its upstream regulators in the Wnt pathway does not lead to GSC loss.

We thank the reviewer for this insightful comment. A likely explanation is that knockdown of individual Wnt ligands or receptors leads to mild phenotypes due to redundancy, whereas Pan/Tcf acts downstream as a context-dependent transcriptional regulator that functions as a repressor in the absence of Wnt signaling and as an activator upon pathway activation; its depletion thus disrupts both repression and activation, resulting in stronger and more penetrant phenotypes. We explain this now in the Discussion a bit more in detail and included the following: *"A likely reason is that knockdown of individual Wnt ligands or receptors produced only mild phenotypes, likely due to functional redundancy. In contrast, knockdown of the downstream effector Pan/Tcf, which integrates both transcriptional activation and repression, produced stronger and more variable effects, highlighting its central role in transducing Wnt inputs."*

POINT-BY-POINT RESPONSE TO THE REVIEWERS:

We would like to thank all the reviewers again for their constructive criticism and important suggestions. Reviewer 1 and 2 were satisfied with the revisions. Reviewer 3 had three minor points, which we addressed in the following manner:

1. Multiple studies have repeatedly demonstrated that ovo mutant does not have discernable phenotypes in male. Can you speculate/discuss on this possible discrepancy?

We added the following paragraph in the manuscript for clarification:

“Notably, while previous studies suggested that ovo is dispensable in males despite its expression in early germ cells of adult testes, our adult-specific knockout reveals a clear phenotype, indicating a functional requirement in the male germline. This finding is consistent with the work of Hayashi et al. (2017), who showed that maternal ovo-B is essential in primordial germ cells of both sexes. The discrepancy likely reflects the use of hypomorphic alleles or RNAi in earlier studies, which may not have fully disrupted ovo function or captured temporally restricted phenotypes. Together, our results suggest that ovo plays a general role not only in germline development but also in the maintenance of germline identity and function during adulthood.”

-They used ‘GSC diameter’ as a phenotypic criterion, but I don’t believe GSC size is an established phenotype that indicates ‘GSC dysfunction’. GSC diameter would be influenced if they are arrested/slowed down in the G1 phase of the cell cycle, but it is hard to imagine all the mutants they examined exhibit the same ‘smaller GSC’ phenotype. Also, based on the images provided, the GSC diameter does not seem to be smaller in mutants (as much as shown in the graph). Can you provide a better rationale why they chose GSC diameter, and what it (small diameter) may mean? Additionally, the unit of GSC diameter cannot be ‘pixels’, as is shown in the figures (Fig 4).

We do agree with the reviewer that GSC diameter is not an established marker of dysfunction, but smaller size could reflect altered growth, cell cycle progression, or stem cell identity. While not definitive, reduced size alongside lower GSC numbers suggests that the cells are indeed affected, possibly through disrupted niche signaling or intrinsic growth regulation. However, as the reviewer pointed out that this is not an established phenotype indicating GSC dysfunction, we removed this parameter from the manuscript.

-Line 117: mention of ‘secondary spermatocytes’: do they really mean ‘haploid germ cells after meiosis I’? (which is the definition of secondary spermatocytes). If so, the transition from spermatogonia to ‘primary spermatocytes’ is not described.

The reviewer is right, this was a typo and should be primary spermatocytes. We have changed this in the manuscript.